# A 23,000-year-old southern Iberian individual links human groups that lived in Western Europe before and after the Last Glacial Maximum

**A list of authors and their affiliations appears at the end of the paper**

Human populations underwent range contractions during the Last Glacial Maximum (LGM) which had lasting and dramatic effects on their genetic variation. The genetic ancestry of individuals associated with the post-LGM Magdalenian technocomplex has been interpreted as being derived from groups associated with the pre-LGM Aurignacian. However, both these ancestries differ from that of central European individuals associated with the chronologically intermediate Gravettian. Thus, the genomic transition from pre- to post-LGM remains unclear also in western Europe, where we lack genomic data associated with the intermediate Solutrean, which spans the height of the LGM. Here we present genome-wide data from sites in Andalusia in southern Spain, including from a Solutrean-associated individual from Cueva del Malalmuerzo, directly dated to ~23,000 cal yr BP. The Malalmuerzo individual carried genetic ancestry that directly connects earlier Aurignacian-associated individuals with post-LGM Magdalenian-associated ancestry in western Europe. This scenario differs from Italy, where individuals associated with the transition from pre- and post-LGM carry different genetic ancestries. This suggests different dynamics in the proposed southern refugia of Ice Age Europe and posits Iberia as a potential refugium for western European pre-LGM ancestry. More, individuals from Cueva Ardales, which were thought to be of Palaeolithic origin, date younger than expected and, together with individuals from the Andalusian sites Caserones and Aguilillas, fall within the genetic variation of the Neolithic, Chalcolithic and Bronze Age individuals from southern Iberia.

The peopling of Europe was marked by human population expansions and contractions associated with major climatic events. Numerous studies indicate a dramatic population contraction in Palaeolithic Europe during the Last Glacial Maximum (LGM, ~26.6–19 cal kyr BP (refs. [1–4])). Human presence in the archaeological record is documented predominantly by artifacts, mainly stone tools assigned to so-called technocomplexes, rather than by skeletal remains, which are rare in the Palaeolithic record.

With the onset of the LGM, a population decline is observed in central Europe and human populations associated with the Gravettian technologies (33–25 cal kyr BP) retreated to southern latitudes, to regions in today's Italy and central/southeastern Europe[1]. In southwestern Europe, a singular Upper Palaeolithic (UP) technocomplex, the Solutrean, emerged in regions of today's southern France and Iberia by ~24–19 cal kyr BP (refs. [5–7]), which coincides in time with the intensive

✉ e-mail: vanessa_villalba@eva.mpg.de; wolfgang_haak@eva.mpg.de

cold peak of the Heinrich 2 Event and following the LGM[8]. The Solutrean is defined by a suite of new lithic technologies and implements, with regionally distinct lithic point types[9–11] interpreted as an adaptation in response to the hard climatic conditions[1] and more generally as a breakdown of the Gravettian technologies. Some scholars have explained the cultural discontinuity by migratory processes, with putative origins in North Africa on the basis of parallels with Aterian lithic assemblages[12–14]. However, the prevailing consensus sees the Solutrean lithic tradition rooted in western European Late Gravettian technologies[15–17], which had undergone cultural drift due to isolation from other groups and the disruption of extended pan-European networks, adaptation to harsh climatic conditions[1] and demographic pressure[18]. Further support for a local development of the Solutrean was seen in the synchronous origin of the new lithic traditions that would culminate in the Solutrean in the French and Iberian territories[16,19].

On the Iberian peninsula, the archaeological record of the Solutrean documents a dense peripheral dispersion on both the Atlantic and Mediterranean sides of the peninsula[7,20], with occasional occupation of the inner plateau[21]. Solutrean traditions between the Mediterranean/southern Portugal and Cantabrian/Pyrenean regions were considered to be a consequence of territorialism which had followed population contractions and limited space[22,23]. In contrast to the preceding Gravettian and the subsequent Magdalenian, when northern Iberia was more densely occupied, the number of sites associated with the Solutrean is roughly equal in both regions, albeit being dispersed more widely in the south[7,20] and suggests a network of interconnected groups within a limited perimeter.

The available genome-wide data from archaeological contexts older than the LGM in western Europe is scarce and does not yet allow a detailed study of the genomic transformation of the UP human groups of this part of the continent (Supplementary Table 1.1). The oldest genome-wide data published so far come from central and eastern Europe, dated back to ~45–40 cal kyr BP corresponding with when the Initial Upper Palaeolithic (IUP) technologies prevailed and the genotyped individuals show a wide variety of ancestry profiles and levels of Neanderthal admixture (for example, Bacho Kiro_IUP from Bulgaria[24], Oase1[25], Muieri from Romania[26] and Zlatý kůň from the Czech Republic[27]). Conversely, the oldest genome-wide data available from western Europe come from an Aurignacian-associated individual Goyet Q116-1 in today's Belgium[26]. Gravettian-associated individuals from pre-LGM central and southern Europe form a genetic cluster (share more ancestry within the group than with individuals outside the group) (Fig. 1), which was named after the oldest individual, here *Věstonice* cluster[26,28], irrespective of the Gravettian industries they were associated with archaeologically. However, so far, no genome-wide data have been published from western European Gravettian-associated individuals (Fig. 1). Pre-LGM Gravettian-associated groups from central Europe (*Věstonice* cluster) differ genetically from post-LGM Magdalenian-associated groups from both central and western Europe (which also form a genetic cluster coined *Goyet Q2*) (refs. [26,29]), whereas the last had received genetic ancestry first found in an Aurignacian-associated individual Goyet Q116-1 from northwestern Europe. This genomic discontinuity between central European Gravettian-associated individuals and western-central European Magdalenian-associated individuals has been explained by the population contractions during LGM[26] and supported by mitochondrial studies, which noted the disappearance of, for example, mitochondrial DNA haplogroup M during the LGM[30].

Following the Bølling/Allerød warming interstadial (14 cal kyr BP), the *Goyet Q2* cluster was replaced by the *Villabruna* cluster in central Europe, named for its oldest Epigravettian-associated individual from northern Italy[26], but which also includes most of the Epipalaeolithic- and Mesolithic-associated groups from central and western Europe, all of which are also known as western hunter-gatherers (WHG)[31]. In this genetic landscape, Iberian hunter-gatherers (HG) stood out as

they retained higher proportions of the *Goyet Q2*-like ancestry during the Epipalaeolithic and Mesolithic periods and thus are often considered separate[29].

Individuals from western Europe who directly date to the LGM period are essential to address the genetic discontinuity between pre-LGM and post-LGM groups described by ref. [26]. To investigate the role of southern European refugia during the LGM, we generated genome-wide data from several Solutrean-associated human remains from Cueva del Malalmuerzo (Moclín, Granada, Spain) (Fig. 1). Cueva del Malalmuerzo is well known for its rock art paintings that are stylistically attributed to the Solutrean. Although Solutrean industries have been found in the cave[32], so far there are no in situ stratigraphic layers directly associated with this technocomplex. The latest archaeological investigations of the cave uncovered several human remains in a small area, which corresponded to an old archaeological profile from previous excavations.

We sampled additional prehistoric human remains from various cave and rock shelter sites in Andalucia, Spain (Supplementary Information 1), with long occupation histories to establish a time transect in southern Iberia from the LGM to the Neolithic periods. After applying quality filters and radiocarbon dating, we were able to analyse one Solutrean-associated individual from Cueva del Malalmuerzo, two EN individuals from Cueva de Ardales and Las Aguilillas and two Chalcolithic (CA) individuals from Cueva de Ardales and Los Caserones. Individual ADS007 from Cueva de Ardales did not provide enough collagen for radiocarbon dating but enough genome-wide information to perform genetic analyses (Supplementary Table 1.2 and Supplementary Information 1). We present the contextualized genomic results in chronological order.

## Results and discussion

To generate genome-wide data with maximum coverage, several DNA extracts and single-stranded non-uracil-DNA-glycosylase (UDG) libraries from each sample (Supplementary Tables 1.2 and 1.3) were prepared following established protocols[33]. Our final dataset ranged from 0.51× to 8.7× average coverage on targeted 1,240,000 single-nucleotide polymorphism (SNP) sites (Supplementary Table 1.2). After 1,240,000 SNP capture the merged libraries underwent a second round of quality control, applying a minimum SNP cutoff for robust ancient DNA authentication and contamination estimation (Supplementary Information 2, Supplementary Figs. 1–3 and Supplementary Tables 1.4–1.9).

### The genomic make-up of Solutrean hunther-gatherers of Cueva del Malalmuerzo

Two human teeth were recovered during an archaeological survey of Cueva del Malalmuerzo (MLZ): MLZ003 (arch ID, MALM16 SUP2.1; tooth 34) and MLZ005 (arch ID, MALM16 Sector A 8.2; tooth 33). Samples MLZ003 and MLZ005 were found to be contemporaneous and radiocarbon dated to a period when the Solutrean technocomplex prevailed (MLZ003, 23,016–22,625 cal yr BP; MLZ005, 22,979–22,570 cal yr BP), concordant with the stylistic rock art found in the cave[34] (Supplementary Table 1.2) and thus present the oldest genomic data from UP human remains in Iberia. We found that both teeth belong to the same individual and thus merged the data (MLZ003005 or MLZ henceforth) for downstream population genetics analyses (Supplementary Information 2, Supplementary Fig. 2a,b and Supplementary Table 1.6). The final average coverage on targeted SNP sites was 0.41×, which corresponds to 226,914 autosomal SNPs in the 1,240,000 panel (Supplementary Table 1.2).

MLZ carries mitochondrial DNA-haplogroup U2'3'4'7'8'9 (Supplementary Tables 1.10 and 1.11). The oldest individual carrying the derived mtDNA-haplogroup U2 is Kostenki14 (~38 cal kyr BP, Russia)[35,36], while the more basal mtDNA-haplogroup U2'3'4'7'8'9 is restricted to individuals in southwestern Europe, with Paglicci108 being the oldest Palaeolithic individual (~27.8 cal kyr BP, Italy)[26], followed by

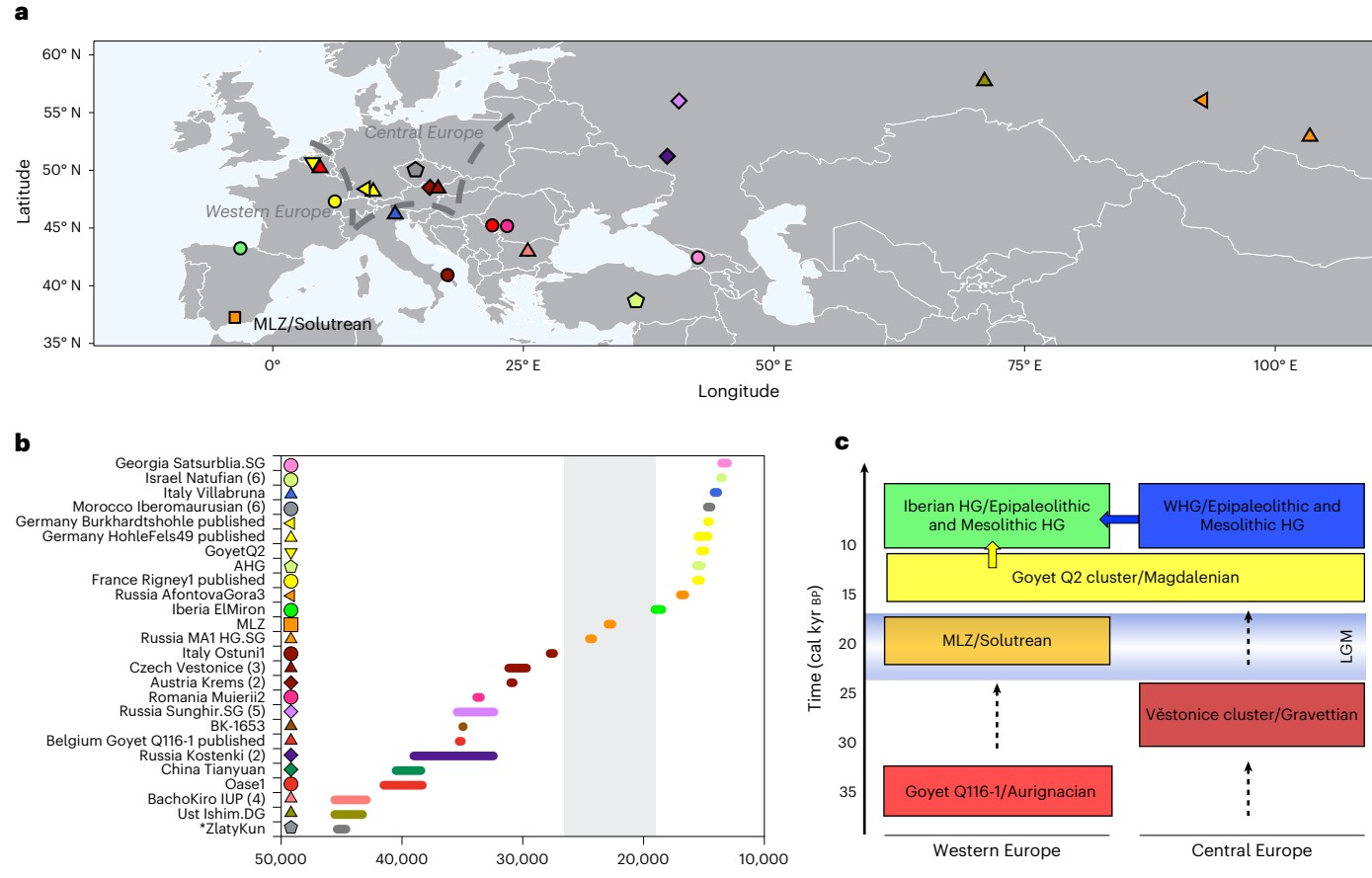

**Fig. 1 | Chronological and geographical overview of newly reported and relevant published individuals. a**, Geographical distribution of Pleistocene individuals with genome-wide data (>20,000 SNPs covered in the 1,240,000 SNP panel; coloured symbols, consistent with individuals and symbols on the *y* axis of **b**). **b**, Chronological distribution of Pleistocene individuals with genome data. The grey bar indicates the extent of the LGM (*Zlatý kůň is dated genetically to -45 kyr BP) (ref. [27]). **c**, Genetic overview of the western and central Europe UP and their correspondence with technocomplexes (where possible). Arrows with dashed lines show gaps in the genetic record. See Supplementary Table 1.1 for a detailed description of the individuals.

Rigney (-15.5 cal kyr BP, France)[26], Oriente_C (-14 cal kyr BP, Sicily)[37], Grotta dell'Uzzo (-10 cal kyr BP, Sicily)[38] and Balma Guilanyà (-13 cal kyr BP, Spain)[29]. The geographic distribution of U2'3'4'7'8'9 is consistent with an early spread of human groups into western Europe and was suggested to have survived the LGM in the Iberian and Apennine refugia[30].

MLZ carried Y chromosome haplogroup C1 (Supplementary Table 1.2), which was also found in individuals from Bacho Kiro IUP (~45 cal kyr BP, Bulgaria). The more basal Y-haplogroup C was found in Paglicci_133 (-33 cal kyr BP, Italy), Cioclovina_1 (-32 cal kyr BP, Romania) and Kostenki_12 (C, -32 cal kyr BP, Russia)[26] and the derived Y-haplogroup C1a2 was found in Goyet Q116-1 (-35 cal kyr BP, Belgium)[26] and Sunghir (-34 cal kyr BP, Russia)[39].

To characterize the genomic profile of MLZ, we estimated genetic similarities among all published Palaeolithic and Mesolithic HGs including the new data using $f_3$-outgroup statistics of the form $f_3(HG1, HG2; Mbuti)$. In the resulting heatmap (Supplementary Information 3 and Supplementary Fig. 4), MLZ clusters with the later Magdalenian-associated individuals from the *Goyet Q2* cluster and some Epipalaeolithic and Mesolithic HGs from Iberia. These results suggest a genetic ancestry that is similar, or related to, the one found to be characteristic for Magdalenian-associated individuals[26] and present in an admixed form in Iberian HGs[29,40].

Multidimensional scaling (MDS) of the transformed pairwise-distance $f_3$-matrix ($1 - f_3$, Fig. 2a) shows that MLZ falls outside the genetic variation of the preceding central European Gravettian-associated individuals (*Věstonice* cluster[26]). Interestingly, MLZ falls between the Aurignacian-associated Goyet Q116-1 and the Magdalenian-associated individuals from the *Goyet Q2* cluster, to the exclusion of El Mirón, who falls within the Iberian HG cline that bridges WHG- and *Goyet Q2*-like ancestries[29]. We then calculated $f_4$-statistics of the form $f_4(GoyetQ2 cluster, GoyetQ116-1; MLZ, Mbuti)$ to test whether Goyet Q116-1 and Magdalenian-associated individuals are cladal with respect to MLZ (Fig. 2b and Supplementary Table 2.1). We find that MLZ shares more genetic drift with Magdalenian-associated individuals than with Goyet Q116-1. However, when testing whether Magdalenian-associated individuals and MLZ are symmetrically related to Goyet Q116-1 using the contrasting $f_4(MLZ, GoyetQ2 cluster; GoyetQ116-1, Mbuti)$, we observe an excess of shared drift between MLZ and Goyet Q116-1, for example when HohleFels49, Goyet Q2 and El Mirón are used as proxies for Magdalenian-associated ancestry (Fig. 2c and Supplementary Table 2.2). These results suggest that MLZ represents a lineage that is genetically intermediate between Goyet Q116-1 and individuals from the *Goyet Q2* cluster. In line with the chronology, Goyet Q116-1 is more genetically similar to MLZ than to the *Goyet Q2* cluster, whereas MLZ is genetically more similar to the *Goyet Q2* cluster than to the preceding Goyet Q116-1 (Fig. 2b,c). Identifying MLZ as a member of a lineage that contributed genetically to Magdalenian-associated individuals is consistent with the archaeological record that postulates an emergence of the Magdalenian technocomplex in regions of northern Iberia and southern France[41,42].

We then explored whether MLZ and Aurignacian-associated Goyet Q116-1 or the later *Goyet Q2* cluster were symmetrically related with respect to the *Věstonice* cluster using $f_4$-statistics of the form $f_4$(*MLZ, GoyetQ2 cluster/GoyetQ116-1; Věstonice cluster, Mbuti)* (Fig. 2d,e and Tables 2.3 and 2.4) but observed no excess shared drift with *Věstonice* cluster individuals. This means that the genetic discontinuity between pre-LGM *Věstonice* cluster and post-LGM *GoyetQ2 cluster*, as reported by ref. [26], was not driven by the harsh climatic change, as the differentiation is already visible in MLZ, who directly dates to the height of the LGM. This implies that at least two genetically distinct groups must have existed in Europe when the Gravettian technocomplex prevailed: one in western Europe, represented by Goyet Q116-1 and a second in central (and perhaps eastern) Europe, described as the *Věstonice* cluster[26]. Our results match technological studies which suggest that the Solutrean was rooted in western Gravettian technologies[15,16,19] and the resemblance of the rock art associated with the Gravettian and Solutrean in western Europe[43]. By contrast, this result renders a monocentric central European origin of the Gravettian unlikely[44].

Given that the Solutrean is restricted to southern France and Iberia and assuming that southwestern Europe was a geographical refugium for UP populations during the LGM, population continuity through time is a parsimonious explanation. However, given the lack of pre-LGM genetic data from Iberia, the presence of *Věstonice*-like ancestry in Iberia before the LGM cannot be ruled out. The last had reached the Italian Peninsula, where it was later replaced by Epigravettian-associated Villabruna-like ancestry[26] and a similar replacement scenario could also be possible for Iberia.

## Signals of deep ancestry

Recent studies have shown that IUP individuals from Bacho Kiro (45 cal kyr BP, Bulgaria), Tianyuan (40 cal kyr BP, China) and Goyet Q116-1 (35 cal kyr BP, Belgium) carried ancestry from an IUP population which had inhabited Eurasia before the split of West Eurasian and East Asian populations[24] (Supplementary Information 5). Using $f_4$-statistics of the form $f_4$(*MLZ, Kostenki14; test, Mbuti)* and Kostenki14 as the baseline for European Palaeolithic ancestry[26] we show that MLZ shares excess genetic drift with Bacho Kiro IUP, Goyet Q116-1 and Tianyuan (Fig. 3a, Supplementary Fig. 5a and Supplementary Table 2.5).

Our results confirm that part of the IUP ancestry present in Bacho Kiro and Tianyuan also survived in Southern Iberia MLZ until 23 cal kyr BP, ~12,000 yr later than the Aurignacian-associated Goyet Q116-1, the youngest previously known individual with traces of this ancestry. Initially, this IUP ancestry was attributed to East Asians as it is present in higher proportion in the Tianyuan individual, who shares more alleles with present-day East Asians than present-day or ancient Europeans[26,45]. The same type of ancestry was also observed in Goyet Q116-1 (ref. [26]), who is more closely related to modern and ancient Europeans but still shares excess affinity to Tianyuan. Others[45] have postulated an early pan-Eurasian population, which predated the split time of Europeans and Asians, as opposed to a back migration from Tianyuan-related Asian groups into Europe after the split. The oldest genomic data available from Bacho Kiro cave support the hypothesis of the existence of an Early Eurasian Bacho Kiro-like population that contributed to Ust' Ishim, Tianyuan, Goyet Q116-1 and now also MLZ, but to the exclusion of other UP populations, including pre-Gravettian-associated Sunghir and Kostenki14, central European Gravettian or Magdalenian-associated individuals (*Goyet Q2* cluster)[24]. Using $f_4$-statistics of the form $f_4$(*test, Kostenki14; Tianyuan, Mbuti)*, we observe that Goyet Q116-1 and MLZ share more genetic ancestry with Tianyuan than Bacho Kiro IUP does (Fig. 3b and Supplementary Table 2.6). We also observe excess shared ancestry between Tianyuan and MLZ/Goyet Q116-1 when we replace Kostenki14 with Bacho Kiro IUP in $f_4$(*MLZ/GoyetQ116-1, Bacho Kiro IUP; Tianyuan, Mbuti)*, which also returns positive $f_4$-statistics in both tests (MLZ, $Z$ = 2.011; Goyet Q116-1, $Z$ = 2.244) (Supplementary Table 2.7). This trend was already

observed for Goyet Q116-1 and was explained by a higher shared ancestry between the latter and Tianyuan[24]. The Tianyuan-related ancestry present in MLZ might be fully inherited from Goyet Q116-1 as both are symmetrically related to Tianyuan and Bacho Kiro IUP individuals as shown by $f_4$-statistics of the form $f_4$(*MLZ, GoyetQ116-1, Tianyuan, Mbuti)* ($Z$ = 0.282) and $f_4$(*MLZ, GoyetQ116-1, Bacho Kiro IUP, Mbuti)* ($Z$ = 0.705) (Supplementary Table 2.7). We also observe a subtle attraction between Ust'Ishim and MLZ by obtaining overall negative values using $f_4$(*test, Ust'Ishim; MLZ, Mbuti)*, which is consistent with an IUP Bacho Kiro-like group contributing ancestry to Ust'Ishim, Tianyuan, Goyet Q116-1 and, more distantly, to MLZ (Supplementary Fig. 5b, Supplementary Table 2.8 and Supplementary Information 3). Similar levels of Tianyuan and Bacho Kiro IUP attraction between MLZ and Goyet Q116-1, and its persistence over time in MLZ, suggest that this type of early Eurasian ancestry survived in a diluted form in western-most Europe. Ultimately, this observation posits a connection between Aurignacian- and Solutrean-associated individuals in western Europe, the IUP in eastern Europe and Tianyuan in the East and that this genetic legacy persisted in Iberia for ~20,000 yr more (MLZ, ~23 cal kyr BP), while in central (and presumably eastern) Europe, it was superseded and already no longer traceable in pre-Gravettian/Gravettian-associated individuals (~30 cal kyr BP). These results suggest genetic continuity from a population broadly associated with the Aurignacian and represented by Goyet Q116-1 to a population associated with the Solutrean and represented by MLZ in western Europe.

In the case of MLZ, we infer that this type of ancestry must have been brought to southern Iberia by individuals associated with the Aurignacian (sensu lato) as the archaeological record only provides evidence of Early UP industries (for example, Châtelperronian) in northern Iberia which is broadly attributed to Late Neanderthals and not modern humans[46]. Entering the peninsula from southern France, archaeological remains securely assigned to the Proto or Early Aurignacian are only found in northern Iberia[47–49]. For the sites Bajondillo near Málaga[50] and Lapa do Picareiro in central Portugal[51] the presence of an Early Aurignacian technocomplex has also been reported but challenged by several scholars[49,52].

## Signals of admixture in MLZ

We also tested whether MLZ carried UP HG-like ancestries that differ from the Aurignacian-associated individual Goyet Q116-1. First, we continued with $f_4$-statistics of the form $f_4$(*MLZ, Kostenki14, test, Mbuti)* to explore the genetic relationship of MLZ to other ancient individuals. We find a subtle, but non-significant, signal of shared drift with individuals that carried Villabruna-like ancestry MLZ ($Z$ = 2.972) which is absent in Goyet Q116 ($Z$ = 1.783) (Supplementary Table 2.10). We could confirm that the amount of Villabruna-like ancestry in MLZ is less than in Magdalenian-associated individuals from the *Goyet Q2* cluster calculating $f_4$(*MLZ/GoyetQ2 cluster, Goyet-Q116-1; Villabruna, Mbuti)* (Supplementary Table 2.9). Here, we obtain significantly positive $f_4$-statistics for all Magdalenian-associated individuals, which indicates a contribution of Villabruna-like ancestry to *Goyet Q2* cluster individuals. Testing MLZ resulted in a non-significant $Z$-score (1.628). However, taking these results into account, a slightly higher amount of Villabruna-like ancestry in MLZ than in Goyet Q116-1 cannot be ruled out (Supplementary Table 2.9).

We also observe consistently negative $f_4$-statistics using $f_4$(*MLZ, GoyetQ2 cluster; Villabruna, Mbuti)* with $Z$-scores ranging from −9.568 (El Mirón) to −0.091 (Hohle Fels), which suggests excess affinity to Villabruna-like ancestry in the *Goyet-Q2* cluster individuals when compared to MLZ, even though the $f_4$-statistics are not always significant ($|Z|$ > 3) (Supplementary Table 2.9). For the Iberian Peninsula, we find that MLZ shares less drift with Villabruna-like individuals than the later Magdalenian-associated El Mirón, which suggests that the incoming Villabruna-like ancestry did not arrive in Iberia during the time when the Solutrean technocomplex prevailed but probably

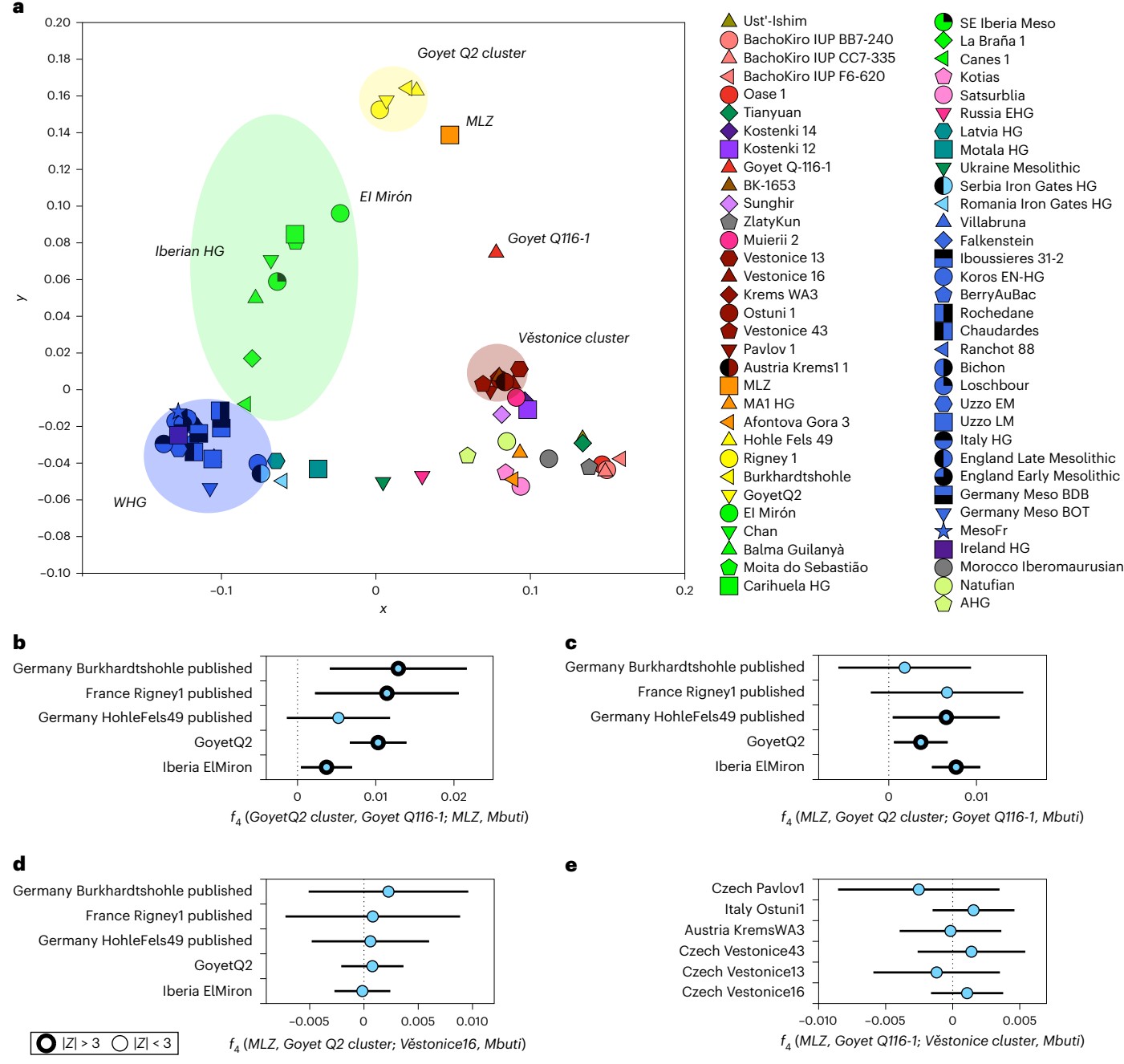

**Fig. 2 | Genetic affinity of the MLZ individual and genetic structure among HGs. a**, MDS plot of the pairwise $f_3$-matrix of the form $f_3(HG1, HG2; Mbuti)$ (Supplementary Fig. 1) transformed into distances using $1 − f_3$. The main genetic clusters mentioned in the paper are highlighted here. **b**, The $f_4$-statistics of the form $f_4(GoyetQ2 cluster, GoyetQ116-1; MLZ, Mbuti)$ show a significant affinity between Magdalenian-associated individuals and MLZ when compared to Goyet Q116-1 (Supplementary Table 2.1). **c**, The $f_4$-statistics of the form $f_4(MLZ, GoyetQ2 cluster; GoyetQ116-1, Mbuti)$ show that MLZ and Magdalenian-associated individuals are not symmetrically related to Goyet Q116-1 and that MLZ shares more genetic drift with Goyet Q116-1 (Supplementary Table 2.2). **d**, The $f_4$-symmetry tests of the form $f_4(MLZ, GoyetQ2 cluster; Věstonice16, Mbuti)$, where

Věstonice 16 is symmetrically related to MLZ and Magdalenian-associated individuals. **e**, The $f_4$-symmetry tests of the form $f_4(MLZ, GoyetQ116-1; Věstonice cluster, Mbuti)$, including other central European, Gravettian-associated individuals, who are symmetrically related to MLZ and Goyet Q116-1. Both tests (**d** and **e**) do not deviate from 0 and thus show that there is no excess of shared drift between MLZ/GoyetQ116-1 and central European, Gravettian-associated individuals (Supplementary Tables 2.3 and 2.4). For all $f_4$-statistics, error bars indicate ±3 s.e. and were calculated using a weighted block jackknife[83] across all autosomes on the 1,240,000 panel (nSNPs = 1,150,639) and a block size of 5 megabases (Mb); |$Z$| > 3 points with thicker outline.

later; or, alternatively, that it had not yet reached the southern part of the peninsula.

Interestingly, MLZ indicates a significantly positive attraction to Natufians ($Z = 3.541$) using $f_4(MLZ, Kostenki14; Natufian, Mbuti)$ (Supplementary Table 2.10). When comparing the different affinities to

Villabruna and Natufian in an extension of the comparable $f_4$-test settings, we observe the following pattern: all post-LGM groups/individuals show significant attraction to Villabruna and Natufian-like ancestry, whereby Villabruna constitutes the type of ancestry which results overall in higher $f_4$-statistics (Fig. 4a and Supplementary Table 2.10). This

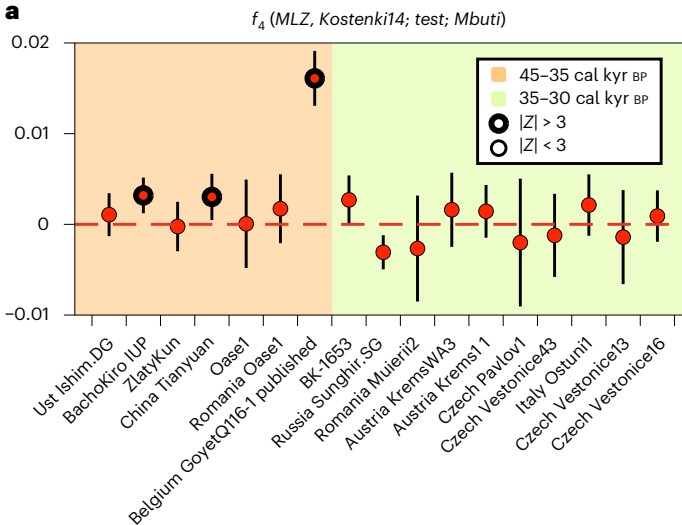

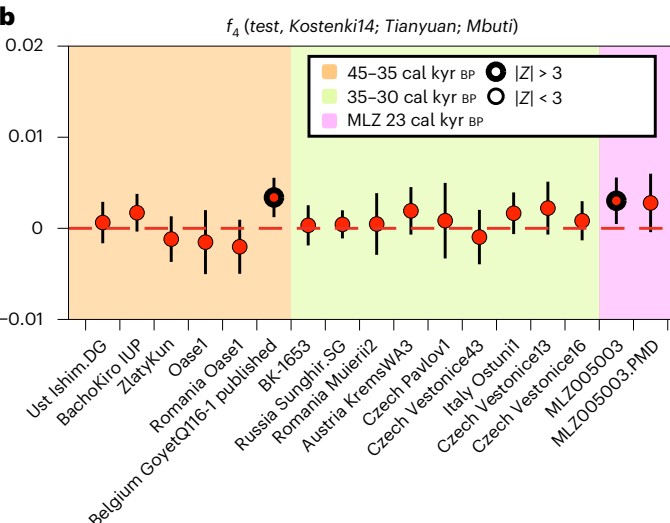

**Fig. 3 | Attraction of MLZ to old UP individuals. a**, The $f_4$-statistics of the form $f_4$(*MLZ*, *Kostenki14*; *test*, *Mbuti*) show significant positive affinity of MLZ to IUP Bacho Kiro, Tianyuan and Aurignacian Goyet Q116-1 (Supplementary Table 2.5). **b**, The $f_4$-statistics of the form $f_4$(*test*, *Kostenki14*; *Tianyuan*, *Mbuti*) show significant positive affinity of MLZ and Goyet Q116-1 to Tianyuan (Supplementary Table 2.6). For all $f_4$-statistics, error bars indicate ± 3 s.e. and were calculated using a weighted block jackknife[83] across all autosomes on the 1,240,000 panel (nSNPs = 1,150,639) and a block size of 5 Mb; |*Z*| > 3 points with thicker outline.

pattern is clearest in well-covered WHG and Magdalenian-associated individuals but a similar trend is seen in the low-coverage individuals from the two groups. Others[26] have already described an increased affinity of Near Eastern populations to the Villabruna/WHG cluster after 14 cal kyr BP and, consequently, suggested a contribution from ancient Near East groups to the *Villabruna* cluster in a southeastern European refugium before, that is, during the LGM or earlier. Villabruna-like ancestry was also detected in the non-basal Eurasian part of the ancestry of Anatolian HG and Natufians, which suggested bidirectionality, that is a contribution of Villabruna-like ancestry to ancient Near Easterners before ~15 cal kyr BP.

Here, we confirm a significant genetic affinity to Villabruna-like ancestry in pre-LGM Gravettian-associated individuals from central Europe but not (or to a much lesser extent) to Natufian-like ancestry. Of note, MLZ is one of the oldest individuals who shows a positive attraction to both Natufians (*Z* = 3.541) and Villabruna (*Z* = 2.972). By contrast, post-LGM populations share substantial ancestry with both

Villabruna and Natufian but more with Villabruna than Natufians, in agreement with ref.[26] (Fig. 4a and Supplementary Table 2.10).

Others[53] have shown that Natufians can be modelled using Villabruna-like ancestry and 'Basal Eurasian' ancestry, which constitutes an inferred population that diverged very early from all non-African populations after the split from African populations[31]. Our results show that MLZ shared an excess of Near Eastern ancestry present in Natufians, which is not explained by Villabruna-like ancestry itself (the oldest WHG with Near Eastern affinity). By contrast, we neither found an indication for Basal Eurasian ancestry in MLZ using other tests (Supplementary Information 6, Supplementary Fig. 6 and Supplementary Table 2.12) nor a high percentage of Neanderthal ancestry (Supplementary Information 7, Supplementary Fig. 7 and Supplementary Table 2.13).

Taken together, we tentatively conclude that the Villabruna-like ancestry present in Natufians and MLZ differs from the one represented by Villabruna. The putative contributing lineage is nonetheless related to Villabruna-like ancestry but carries a higher proportion of Near Eastern ancestry and is present in an admixed form in Natufians.

Finally, we attempted to model these genetic events by performing a reconstruction of the phylogenetic position of MLZ individual using qpGraph (Supplementary Information 8). We found statistical support for a model where MLZ represents a mixture of a population that shared a common ancestor with Goyet Q116-1 (84%) and a population that is ancestral to Villabruna and the clade of all WHG (16%). Magdalenian-associated ancestry represented by El Mirón was found to be a mixture of ancestry similar to MLZ and ancestry similar to Villabruna (Supplementary Figs. 8 and 9).

### Testing for North African ancestry

Given the southernmost location of MLZ in southwest Europe, only ~300 km across from the North African coast and the confirmation of Near Eastern ancestry, we explored a potential trans-Gibraltar connection using $f_4$(*MLZ*, *GoyetQ116-1*; *test*, *Chimp*), where we compare MLZ and Goyet Q116-1 directly with Morocco Iberomaurusian, Natufian and Villabruna as test populations (Fig. 4b and Supplementary Table 2.14). All resulting $f_4$-statistics are positive, indicating a higher affinity to Near Eastern ancestry in MLZ when compared to Goyet Q116-1. Among the three tests, Natufian (*Z* = 2.74) and Morocco Iberomaurusian (*Z* = 2.46) share more genetic drift with MLZ than Goyet Q116-1. We observe a non-significant shift when comparing Morocco Iberomaurusian with Villabruna or Natufian, which suggests the absence of a direct contribution from Morocco Iberomaurusian-like ancestry (who also carry Sub-Saharan-like ancestry) to MLZ (Fig. 4b and Supplementary Table 2.14). However, the Near Eastern-like ancestry represented by Natufians, which is also the major component of Morocco Iberomaurusian, could have spread on both sides of the Mediterranean, where it (later) mixed with Sub-Saharan ancestry in Africa (as seen in Morocco Iberomaurusian) and with Villabruna-like ancestry on the European side of the Mediterranean.

### The genomic legacy of Solutrean-associated HGs in later periods

We next explored whether the genetic legacy of MLZ was still detectable in Holocene HGs from Iberia. Traces of *Goyet Q2*-like ancestry were shown to be present at higher proportions in southern Iberia than in the North, where in turn the proportion of WHG/Villabruna-like ancestry was higher[29].

Applying the same $f_4$(*test*, *Kostenki14*; *Tianyuan*, *Mbuti*) as in Fig. 3, we observed a positive deviation from zero when the Mesolithic individual Moita do Sebastião was tested (*Z* = 2.783), which suggests a subtle affinity to East Asian Tianyuan similar to the one found in MLZ and which, consequently, argues for a persistence of this ancestry in southern Portugal since the UP (Fig. 5a, Supplementary Table 2.6 and Supplementary Information 5). Moita do Sebastião also shows the highest affinity to Tianyuan when all European Mesolithic HGs were

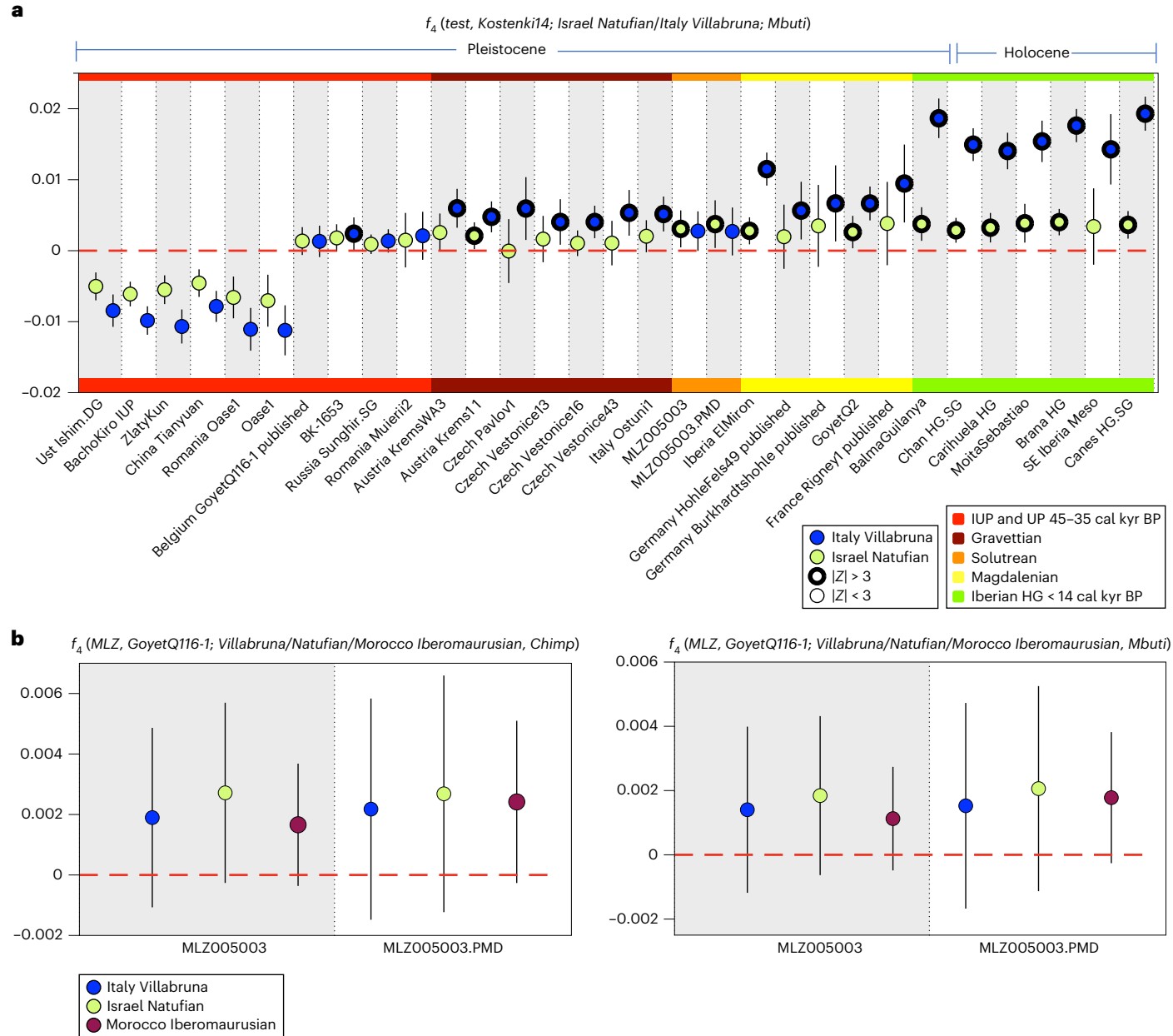

**Fig. 4 | Testing for the presence of Near Eastern ancestry in MLZ. a**, Differential genetic affinity of Pleistocene European HG and Iberian Holocene HG with Natufian and Villabruna-like ancestry calculated using $f_4$(*test, Kostenki14; Natufian, Mbuti*) shown as green circles, and $f_4$(*test, Kostenki14; Villabruna, Mbuti*) shown as blue circles. Bold outlines indicate significant $f_4$-statistics $|Z| > 3$ (Supplementary Table 2.10). Results show that MLZ is the only Pleistocene HG that indicated a significant and equal attraction to both Natufian and Villabruna-like ancestries. **b**, The $f_4$-statistics of the form $f_4$(*MLZ, GoyetQ116-1; Villabruna/Natufian/Morocco Iberomaurusian, Chimp/Mbuti*), used to identify the group that shared most genetic drift with the MLZ lineage (Supplementary Table 2.14). For all $f_4$-statistics, error bars indicate ± 3 s.e. and were calculated using a weighted block jackknife[83] across all autosomes on the 1,240,000 panel (nSNPs = 1,150,639) and a block size of 5 Mb; $|Z| > 3$ points with thicker outline. ka, thousand years ago.

tested in an outgroup-$f_3$(*Mesolithic, Tianyuan; Mbuti*) (Fig. 5b, Supplementary Table 2.15 and Supplementary Information 5). Currently, the strongest affinity to Tianyuan in Holocene European HGs was reported for Eastern European HGs (EHG). This is because the ancestry found in Mal'ta and Afontova Gora individuals (Ancient North Eurasian ancestry) received ancestry from UP East Asian/Southeast Asian populations[54], who then contributed substantially to EHG[55]. However, observing early Asian ancestry in Mesolithic Portugal and EHGs from Russia, at geographically opposite corners of west Eurasia, but not in central Europe, rules out the possibility that ancestry similar to Tianyuan was transmitted through the EHG–WHG admixture cline observed in Mesolithic Europe[56] (Fig. 5b). On the contrary, this result supports the idea of genetic continuity from (at least) the LGM to the Mesolithic in southern Iberia, while other pre- and post-LGM population expansions diluted much of the subtle signal in most other parts of Europe.

We also looked for evidence of a gene flow from North Africa to southern Iberia. We tested for Morocco Iberomaurusian-like ancestry in HGs using $f_4$-statistics of the form $f_4$(*test, Kostenki14; Morocco Iberomaurusian, Mbuti*) (Supplementary Fig. 10a and Supplementary Table 2.16). We find that all populations with WHG (or Near Eastern) ancestry returned positive $f_4$-statistics, which attests to the shared Near Eastern ancestry common to HG groups on both sides of the western

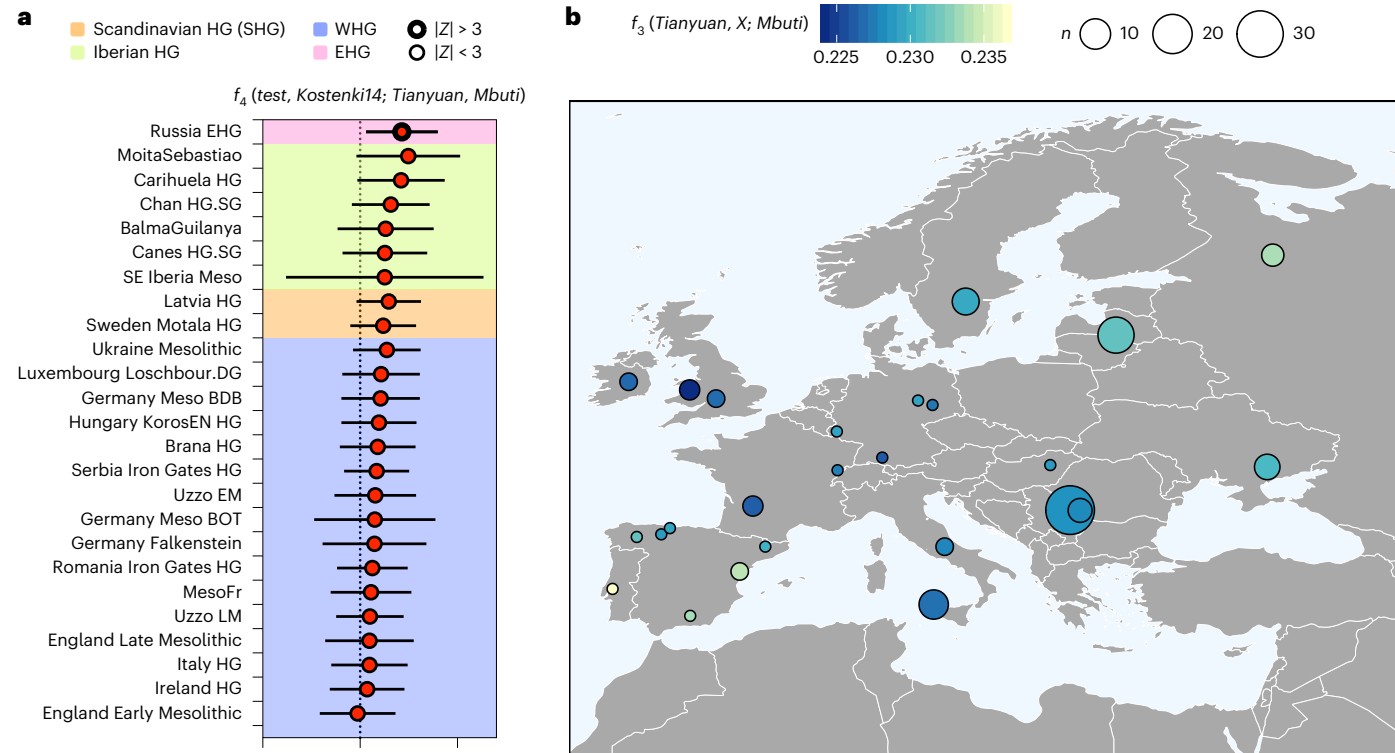

**Fig. 5 | Traces of deep IUP ancestry in Holocene HGs. a**, The $f_4$-statistics of the form $f_4$(*test*, *Kostenki14*; *Tianyuan*, *Mbuti*) show affinity of Moita do Sebastião (non-significant) and EHGs (significant) to Tianyuan (Supplementary Table 2.6). Error bars indicate ± 3 s.e. and were calculated using a weighted block jackknife[83] across all autosomes on the 1,240,000 panel (nSNPs = 1,150,639) and a block size of 5 Mb; |Z| > 3 points with thicker outline. **b**, The $f_3$-outgroup statistics of the form $f_3$(*Tianyuan*, *test*; *Mbuti*) measuring the shared genetic drift between the test population and Tianyuan, highlighting Moita do Sebastião as the Mesolithic Iberian HG with highest shared genetic drift with Tianyuan, indicative of genetic continuity from the Solutrean period in Southern Iberia. Similar $f_3$-outgroup statistics results were obtained for EHGs (Supplementary Table 2.15).

Mediterranean and suggests genetic continuity in southern Iberia. However, when we tested specifically for the Sub-Saharan component of Morocco Iberomaurusian ancestry in MLZ and Moita do Sebastião using $f_4$(*Morocco Iberomaurusian*, *Natufian*; *test*, *Chimp*) we obtained negative results, indicative of excess affinity to the Near Eastern but not to Sub-Saharan ancestry (Supplementary Fig. 10b, Supplementary Table 2.17 and Supplementary Information 9).

With the expansion of farming practices during the Early Neolithic (EN) period, a new form of ancestry reached Iberia[55,57]. These farming groups also carried a small proportion of HG-like ancestry due to local admixture processes along the routes of expansion[56]. In Iberia, we showed that it is possible to track the dual ancestry contributions of Villabruna/WHG-like ancestry and *Goyet Q2*-like ancestry in Neolithic and Copper Age (CA) individuals associated with farming practices[29,58]. To explore the potential genetic legacy of MLZ-like ancestry in later time periods, with a particular focus on southern Iberia, we report and co-analyse two new EN individuals from Cueva de Ardales and Necrópolis de las Aguilillas and two CA individuals from Necrópolis de los Caserones and Cueva de Ardales and one individual from Cueva de Ardales that post-dated the CA period (Supplementary Table 1.2), together with published data from Iberia[29,40,58–60]. EN individuals from Cueva de Ardales (ADS005) and Las Aguilillas (AGS001) cluster in principal component analysis (PCA) space with other Neolithic individuals from Iberia and France. With the exception of Murciélagos, the southern Neolithic forms a separate subcluster within the Iberian EN cluster, with a slight shift upwards on PC2 and left on PC1 towards Iberian HGs (Supplementary Fig. 11a and Supplementary Information 10). Consequently, we grouped all individuals as Southern_Iberia_EN (excluding Murcielagos). After ruling out contributions of North African ancestry in Southern_Iberia_EN (Supplementary Fig. 11b,c,

Supplementary Table 2.18 and Supplementary Information 10), we used qpAdm to model distal sources of genetic ancestry. We tested several combinations of two- and three-way models (Anatolia Neolithic + WHG or Anatolia Neolithic + WHG + either Iran_N, MLZ or Jordan PPNB) with the aim of characterizing the potential additional source(s) of ancestry in Southern_Iberian_EN. We focussed on the different HG ancestries present in the region (for example, MLZ or Goyet Q2-like components) and potentially different Neolithic ancestries, such as Jordan_PPNB or Iran_N-like ancestries, which have been described in some EN groups from the Mediterranean[61]. We successfully modelled the HG ancestry in Southern_Iberia_EN with WHG and MLZ-like ancestry and show that the HG component is larger than in Northern_Iberia_EN individuals, in agreement with previously published data[29,58]. Models with temporally and geographically proximal sources were also supported. However, we cannot distinguish further between MLZ, El Mirón or Moita do Sebastião-like HG ancestries, illustrating the limits of data resolution (Supplementary Fig. 11d and Supplementary Table 2.19). The higher amount of HG ancestry and the Solutrean/Magdalenian-associated genetic legacy suggest a much closer genetic interaction between HGs and farmers in the southern Iberia, perhaps as a result of an earlier spread of farmers (longer co-existence) or more and stronger admixture pulses than in northern Iberia.

Chalcolithic individuals from Necrópolis de los Caserones (CRS002) and Cueva de Ardales (ADS008) cluster with other Southern Iberian CA populations (Supplementary Fig. 11a). The position of ADS007 from Cueva de Ardales in PCA space posits the presence of 'steppe-related ancestry', which was confirmed by several tests applied. Additionally, a non-local status can be suggested for this individual (Supplementary Information 10 and Supplementary Tables 2.20–2.22).

## Conclusions

Genome-wide data from the first Solutrean-associated individual MLZ from Andalucia revealed traces of ancestry from an IUP population that predates the genetic split between European and Asian populations but is still traceable in southern Iberia ~23,000 years ago. This genetic ancestry was also found in the Aurignacian-associated individual Goyet Q116-1, Bacho Kiro and Tianyuan, respectively.

We can also show that pre-LGM (*Věstonice*-like ancestry) and post-LGM (*Goyet Q2*-like ancestry) groups were separated during the LGM as we find no substantial traces of *Věstonice*-like ancestry contribution in southern Iberia. This suggests a scenario in which Gravettian-associated individuals in western Europe were genetically distinct from those in central Europe. It is also possible that *Věstonice*-like ancestry in southern Iberia had been replaced when populations retreated further south during the height of the LGM, or had not initially reached the most southernmost parts of the Iberian Peninsula. Individual MLZ also carried ancestry that is shared with Near Eastern Natufian-associated individuals, confirming the presence of this ancestry in Europe before the LGM. The MLZ lineage contributed substantially to post-LGM Magdalenian-associated individuals, which attests to genetic continuity in western Europe that spans the LGM. While more complex scenarios are possible, the observed genetic continuity suggests that the Iberian Peninsula, as a 'southern genetic refugium', could have sustained a stable population before, during and after the LGM, with no evidence for significant population turnover events but followed by an early and substantial contribution of Villabruna-like HG ancestry soon after. However, this changed profoundly with the arrival of EN farmers, who brought new ancestry from western Anatolia and the Near East. Southern Iberia in particular retained a higher proportion of HG ancestry related to Solutrean- and Magdalenian-associated individuals than other regions of the peninsula. We refer readers to ref. [62], which reports new genomic data from 116 HG individuals from Palaeolithic to Mesolithic Europe.

## Methods

### Direct AMS [14]C

We performed radiocarbon dating on the same skeletal element used for aDNA analysis following refs. [63–66]. Collagen extraction was performed in the Department of Human Evolution at the Max Planck Institute for Evolutionary Anthropology, Leipzig, Germany, and [14]C measurements were done in Curt-Engelhorn-Zentrum Archäometrie gGmbH, Mannheim. Calibrations were performed using OxCal v.4.4 (ref. [67]) and the IntCal 20 curve[68] (Supplementary Table 1.2).

### aDNA analysis

We extracted and prepared DNA for next-generation sequencing in two different dedicated aDNA facilities (Jena and Leipzig). All the 16 human samples were teeth. When possible, after cleaning, we split the crown and root by a hand saw and drilled both from inside. Protocols used for aDNA extraction and non-UDG-treated, single-stranded library preparation are described in ref. [33]. Following the quality assessment (percentage of human DNA, characteristic aDNA damage, percentage of complexity/clonality) of the initial shotgun sequencing data from several DNA libraries from each of the 16 individuals sampled, 8 libraries with >0.08% endogenous DNA or higher were subsequently enriched for ~1,240,000 SNPs[69], 26 libraries from 12 individuals were enriched for the complete mitogenome[25,70] and 18 libraries from four males for mappable regions of the non-recombining parts of the Y chromosome[71] using independent DNA–DNA hybridization capture techniques. Following DNA capture, libraries were sequenced for 20–40 million reads using single end configuration (1 × 75 base pair (bp) reads) on a Illumina HiSeq4000 platform. The success rate of 50% is explained by the challenging climatic conditions and that we could not select petrous bones that were shown to yield a high amount of endogenous human DNA such as refs. [72,73] but were limited to the few scattered skeletal elements.

Sequence data were demultiplexed on the basis of specific pairs of indexes and processed with the EAGER (1.92.59) pipeline (Supplementary Table 1.3). The pipeline includes adaptor removal (v.2.3.1)[74], mapping against the Human Reference Genome hs37d5 with BWA (v.0.7.12)[75] using aln and samse commands (-l16500, -n0.01, -q30) and removing duplicates with DeDup (v.0.12.2)[76]. Genotypes were called using pileupcaller (https://github.com/stschiff/sequenceTools) and the flag -singlestrandedmode, which entirely removed the aDNA damage.

### aDNA authentication and quality controls

We first used MapDamage (v.2.0.6)[77] to determine the deamination rate at both ends of the sequencing reads. We observed damage patterns consistent with non-UDG treatment in most cases (Supplementary Fig. 1a). We used sex determination via the rescaled X ratio versus Y ratio scatter plot as quality control for contamination from the opposite sex of the individual analysed (Supplementary Fig. 1b,c). For a non-contaminated library we expect an X ratio of ~1 and a Y ratio of ~0 for females and an X and Y ratio of ~0.5 for males. We also applied PMD filtering tools[78] to evaluate the deviation from the non-PMD-filtered version of the individual on a PCA (Supplementary Fig. 2), as well as the replication of all the tests shown in this work using the PMD-filtered version too. Finally, we applied a quantitative method to estimate nuclear contamination in males using the ANGSD[79] which estimates heterozygosity at polymorphic sites on the X chromosome in males (Supplementary Table 1.8). The last quantitative method applied to estimate contamination in genetic males and females was the estimation of mitochondrial contamination using ContamMix[80], which quantifies the heterozygosity on the individual mitochondrial reads with a comparative mitochondrial dataset of 311 global mitogenomes (Supplementary Table 1.9).

### Biological relatedness

We calculated the pairwise-mismatch rate (PMR)[81] between ancient individuals based on pseudo-haploid 1,240,000 SNP capture data. To estimate the degree of relatedness among MLZ libraries, we calculated the baseline PMR for identical samples/twins using several libraries from our MLZ individuals and we compared the mean value with the merged MLZ003 versus the merged MLZ005 PMR value. On the basis of the PMR results we concluded that samples MLZ003 and MLZ005 were from the same individual (Supplementary Table 1.6). We replicated the process separately for new EN and CA individuals from Cueva de Ardales and Necrópolis de las Aguillas.

To further validate the finding that MZ003 and MLZ005 are the same individual, when we lacked a robust estimate of the background relatedness, we performed the following test: we calculated the PMR for all pairs of libraries for MLZ003 and MLZ005, which yielded three categories of PMRs—comparisons between MLZ003 libraries (MLZ003/MLZ003), comparisons between MLZ005 libraries (MLZ005/MLZ005) and comparisons between MLZ003 and MLZ005 libraries (MLZ003/MLZ005). Leveraging the fact that the ranks of the PMR values will be invariant to normalization for background relatedness, we used non-parametric, pairwise Wilcoxon rank sum tests to evaluate if there was a significant difference in distribution for the PMR values from the three categories. We found no such difference (adjusted $P > 0.15$), even when we restricted PMR values from pairs with >500 overlapping SNPs ($P = 0.1974$). Since we find no statistically significant difference between the distributions of the PMR values for combinations of libraries of MLZ003/MLZ003, MLZ003/MLZ005 and MLZ005/MLZ005, we have no evidence to suggest that MLZ003 and MLZ005 are not the same individual.

### Datasets

We merged the newly reported ancient data with data from the Allen Ancient DNA Resource (v.37.2; https://reich.hms.harvard.edu) plus the newly published data from refs. [24,27,28,38]. We generated two datasets,

the 1,240,000 genotype set used for all the statistics presented in the paper and one with the intersected SNPs from the Human Origins panel (~600,000 SNPs) used to perform PCA analyses. In total, 179 Palaeolithic and Mesolithic individuals, all covered at >20,000 SNPs, were used in this study.

## Population genetic analysis

**Principal component analysis.** We used the smartpca software from the EIGENSOFT package (v.6.0.1)[82] with the present-day groups (global PCA), present-day west Eurasian groups (West Eurasian PCA) and present-day west Eurasian, North African and Sub-Saharan groups (West Eurasians–North Africans–Subsaharans). The ancient individuals were projected onto the PCA scaffold which was calculated using the modern individuals using the parameters 'lsqproject:YES' and 'shrinkmode:YES'. We project the genotyped PMD-filtered version and the non-PMD-filtered version to evaluate potential signs of contamination[78]. A global PCA was used to infer East Asian ancestry and by adding North Africans and Sub-Saharan populations to the West Eurasian PCA we could infer potential African ancestry.

**F-statistics.** F-statistics were calculated with qpDstat from ADMIX-TOOLS (https://github.com/DReichLab). The $f_3$-outgroup statistics were used to calculate the affinity matrix with all the possible HG pairwise combinations. The $f_4$-statistics were used to test for cladality and admixture. Standard errors were calculated with the default block jackknife and all plots display 3 s.e.

**Multidimensional scaling analysis.** We applied MDS using the R package cmdscale. Euclidean distances were computed with the genetic distances calculated from the $f_3$-outgroup matrix on the form $1 - f_3$ pairwise values among all possible HG pairwise combinations following ref. [26].

**qpGraph.** We used qpGraph from ADMIXTOOLS (https://github.com/DReichLab) to construct the phylogeny of our Palaeolithic individual from Cueva del Malalmuerzo. We built the hypothetical topology of the tree based on the previous statistical analysis ($f_3$- and $f_4$-statistics) and used qpGraph to clarify the order of the genetic events that were inferred previously. The trees were built by order of complexity and the ones with the difference between the observed and fitted f-statistics |Z| > 3 were rejected. We also excluded models with 0% ancestry stream estimates. We used the options 'outpop: NULL' rather than specifying an outgroup population, 'useallsnps: YES', 'lambdascale: 1' and 'diag: 0.0001' as used in ref. [54].

**qpWave and qpAdm.** To estimate admixture proportions we used the qpWave and qpAdm programs from the ADMIXTOOLS v.5.1 package (https://github.com/DReichLab), with the 'allsnps: YES' option. With qpWave, we evaluate the number of sources needed to model our target population. With qpAdm, we quantified the proportion of genetic ancestry contributed by each source in the target population. Programs qpWave and qpAdm were used in post-Palaeolithic populations co-analysed in this study.

**Mitochondrial haplogroup assignment.** We extracted reads from the mitocapture that mapped exclusively to the mitochondrial reference and built consensus sequences using sites which had been covered by a minimum of two reads and had a minimum allele frequency of 0.1. Consensus sequences were uploaded to HaploGrep2 v.2.1.1 (available via https://haplogrep.uibk.ac.at/) for an automated mitochondrial haplogroup assignment based on phylotree (mtDNA tree build 17, available via http://www.phylotree.org/) (Supplementary Table 1.11).

**Y-haplogroup assignment.** We genotyped the Y chromosome reads using a Y-SNP list from the ISOGG (International Society of Genetic Genealogy v.15.73) dataset included in the 1,240,000 and the in-house

Y-capture probes using the procedure described in ref. [71] (Supplementary Table 1.2).

## Reporting summary

Further information on research design is available in the Nature Portfolio Reporting Summary linked to this article.

## Data availability

Genomic data (BAM format) are available at the European Nucleotide Archive under accession number PRJEB58642.

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

## Acknowledgements

We thank all members of the Archaeogenetics Department of the Max Planck Institute for Evolutionary Anthropology, especially the Population Genetics group, and we thank Paleogenomics Lab members from Instituto de Biología Evolutiva for the data discussion. We especially thank M. Hajdinjak and I. Olalde for valuable comments and suggestions. We thank G. Brandt, A. Wissgott, S. Clayton and K. Prüfer for sequencing and handling the raw data. We thank all the members of the general research project in Cueva de Ardales and Sima de las Palomas de Teba. We thank all colleagues, archaeologists, geologists and experts in different scientific disciplines, as well as students from the Universities of Cádiz, Cologne and Tübingen, who have participated in the field and laboratory works. We are grateful for the great collaboration of the Town Councils of Ardales and Teba, within the framework of collaboration agreements with the University of Cadiz and the Neanderthal Museum. This work was supported by the Max Planck Society (V.V.-M., A.B.R., J.K. and W.H.), Margarita Salas 2022 funded by Unión Europea-Next Generation EU (V.V.-M.) and the European Research Council under the European Union's Horizon 2020 research and innovation programme (grant agreement no. 771234-PALEoRIDER to W.H. and no. 803147-RESOLUTION to S.T.). The information provided on recent dating and archaeological records from Cueva de Ardales is part of the General Research Project, authorized by the Consejería de Cultura y Patrimonio Histórico de la Junta de Andalucía: entitled 'Prehistoric societies (from the Middle Palaeolithic to the Final Neolithic) in the Cueva de Ardales and Sima de las Palomas de Teba (Málaga, Spain)'. A geoarchaeological, chronological and environmental study was conducted under the direction of J.R.-M. and G.-C.W. between 2015 and 2021. Excavations and field and laboratory studies have been carried out in Cueva de Ardales within the framework of the projects entitled 'Analysis of prehistoric societies from the Middle Palaeolithic to the Late Neolithic on the two shores of the Strait of Gibraltar'. Relaciones y contactos—HAR2017-8734P (Ministerio de Economía, Industria y Competitividad-Agencia Estatal de Investigación, cofinanced by FEDER funds) for which J.R.-M. and Salvador Domínguez-Bella are the responsible researchers. Collaborative Research Centre—CRC 806 Our Way to Europe, funded by the Deutsche Forschungsgemeinschaft (DFG-German Research Foundation). Project no. 57444011, under the responsibility of G.-C.W.

## Author contributions

V.V.-M., G.-C.W. and W.H. conceived the study. M.S.v.d.L., H.F., S.T. and F.A. performed or supervised laboratory work. V.V.-M. and A.B.R. performed genetic analysis. L.C., P.C.D., J.R.-M., G.-C.W. and W.H. assembled samples and/or provided archaeological context. W.H. provided guidance in data analysis and H.Y., C.L.-F., C.P. and J.K. provided feedback in data analysis. V.V.-M. and W.H. wrote the paper with input from all authors.

## Funding

## Competing interests

The authors declare no competing interests.

## Additional information

**Correspondence and requests for materials** should be addressed to Vanessa Villalba-Mouco or Wolfgang Haak.

**Vanessa Villalba-Mouco** [1,2,3] ✉, **Marieke S. van de Loosdrecht**[4,5], **Adam B. Rohrlach** [1,6], **Helen Fewlass**[7], **Sahra Talamo**[7,8], **He Yu** [1,9], **Franziska Aron**[4], **Carles Lalueza-Fox** [3,10], **Lidia Cabello**[11], **Pedro Cantalejo Duarte**[12], **José Ramos-Muñoz** [13], **Cosimo Posth** [14,15,1], **Johannes Krause** [1], **Gerd-Christian Weniger**[16] & **Wolfgang Haak** [1] ✉

[1]Department of Archaeogenetics, Max Planck Institute for Evolutionary Anthropology, Leipzig, Germany. [2]Instituto Universitario de Investigación en Ciencias Ambientales de Aragón, IUCA-Aragosaurus, Zaragoza, Spain. [3]Institute of Evolutionary Biology, CSIC-Universitat Pompeu Fabra, Barcelona, Spain. [4]Department of Archaeogenetics, Max Planck Institute for the Science of Human History, Jena, Germany. [5]Biosystematics Group, Wageningen University, Wageningen, the Netherlands. [6]School of Mathematical Sciences, University of Adelaide, Adelaide, South Australia, Australia. [7]Department of Human Evolution, Max Planck Institute for Evolutionary Anthropology, Leipzig, Germany. [8]Department of Chemistry G. Ciamician, Alma Mater Studiorum, University of Bologna, Bologna, Italy. [9]State Key Laboratory of Protein and Plant Gene Research, School of Life Sciences, Peking University, Beijing, China. [10]Natural Sciences Museum of Barcelona (MCNB), Barcelona, Spain. [11]University of Málaga and Grupo HUM-440 University of Cádiz, Cádiz, Spain. [12]Investigador senior cuevas de Ardales y Malalmuerzo, Málaga, Spain. [13]Departamento de Historia, Geografía y Filosofía, Universidad de Cádiz, Cádiz, Spain. [14]Institute for Archaeological Sciences, Archaeo- and Palaeogenetics, University of Tübingen, Tübingen, Germany. [15]Senckenberg Centre for Human Evolution and Palaeoenvironment, University of Tübingen, Tübingen, Germany. [16]Institute of Prehistoric Archaeology, University of Cologne, Cologne, Germany. ✉e-mail: vanessa_villalba@eva.mpg.de; wolfgang_haak@eva.mpg.de

Wolfgang Haak

# Reporting Summary

## Statistics

For all statistical analyses, confirm that the following items are present in the figure legend, table legend, main text, or Methods section.

| n/a | Confirmed | |
|---|---|---|
| ☐ | ☒ | The exact sample size (*n*) for each experimental group/condition, given as a discrete number and unit of measurement |
| ☐ | ☒ | A statement on whether measurements were taken from distinct samples or whether the same sample was measured repeatedly |
| ☐ | ☒ | The statistical test(s) used AND whether they are one- or two-sided *Only common tests should be described solely by name; describe more complex techniques in the Methods section.* |
| ☒ | ☐ | A description of all covariates tested |
| ☐ | ☒ | A description of any assumptions or corrections, such as tests of normality and adjustment for multiple comparisons |
| ☐ | ☒ | A full description of the statistical parameters including central tendency (e.g. means) or other basic estimates (e.g. regression coefficient) AND variation (e.g. standard deviation) or associated estimates of uncertainty (e.g. confidence intervals) |
| ☐ | ☒ | For null hypothesis testing, the test statistic (e.g. *F*, *t*, *r*) with confidence intervals, effect sizes, degrees of freedom and *P* value noted *Give P values as exact values whenever suitable.* |
| ☒ | ☐ | For Bayesian analysis, information on the choice of priors and Markov chain Monte Carlo settings |
| ☒ | ☐ | For hierarchical and complex designs, identification of the appropriate level for tests and full reporting of outcomes |
| ☒ | ☐ | Estimates of effect sizes (e.g. Cohen's *d*, Pearson's *r*), indicating how they were calculated |

*Our web collection on statistics for biologists contains articles on many of the points above.*

## Software and code

Policy information about availability of computer code

| Data collection | fastqc (0.11.4)<br>EAGER (1.92.59)<br>AdapterRemoval v2.3.1<br>BWA (v0.7.12)<br>DeDup (v0.12.2)<br>MapDamage (v2.0.6)<br>samtools (v1.3)<br>pileupCaller v8.6.5 |
|---|---|
| Data analysis | ANGSD (0.910)<br>ContamMix<br>ADMIXTOOLS (5.1) ( qp3Pop, qpDstats, qpGraph, qpF4Ratio)<br>Haplogrep 2 (v2.4.0)<br>R version 3.6.2 |

For manuscripts utilizing custom algorithms or software that are central to the research but not yet described in published literature, software must be made available to editors and reviewers. We strongly encourage code deposition in a community repository (e.g. GitHub). See the Nature Portfolio guidelines for submitting code & software for further information.

## Data

Policy information about availability of data

All manuscripts must include a data availability statement. This statement should provide the following information, where applicable:
- Accession codes, unique identifiers, or web links for publicly available datasets
- A description of any restrictions on data availability
- For clinical datasets or third party data, please ensure that the statement adheres to our policy

> Genomic data (BAM format) are available at the European Nucleotide Archive under accession number PRJEB58642.

## Human research participants

Policy information about studies involving human research participants and Sex and Gender in Research.

| | |
|---|---|
| Reporting on sex and gender | *Use the terms sex (biological attribute) and gender (shaped by social and cultural circumstances) carefully in order to avoid confusing both terms. Indicate if findings apply to only one sex or gender; describe whether sex and gender were considered in study design whether sex and/or gender was determined based on self-reporting or assigned and methods used. Provide in the source data disaggregated sex and gender data where this information has been collected, and consent has been obtained for sharing of individual-level data; provide overall numbers in this Reporting Summary. Please state if this information has not been collected. Report sex- and gender-based analyses where performed, justify reasons for lack of sex- and gender-based analysis.* |
| Population characteristics | *Describe the covariate-relevant population characteristics of the human research participants (e.g. age, genotypic information, past and current diagnosis and treatment categories). If you filled out the behavioural & social sciences study design questions and have nothing to add here, write "See above."* |
| Recruitment | *Describe how participants were recruited. Outline any potential self-selection bias or other biases that may be present and how these are likely to impact results.* |
| Ethics oversight | *Identify the organization(s) that approved the study protocol.* |

Note that full information on the approval of the study protocol must also be provided in the manuscript.

# Field-specific reporting

Please select the one below that is the best fit for your research. If you are not sure, read the appropriate sections before making your selection.

☒ Life sciences  ☐ Behavioural & social sciences  ☐ Ecological, evolutionary & environmental sciences

For a reference copy of the document with all sections, see nature.com/documents/nr-reporting-summary-flat.pdf

# Life sciences study design

All studies must disclose on these points even when the disclosure is negative.

| | |
|---|---|
| Sample size | No statistical methods were used to determine ancient DNA sample size a priori. |
| Data exclusions | Data from specimens that showed insufficient levels of ancient DNA content or high levels of DNA contamination were excluded for further analyses. |
| Replication | Replication is achieved by comparing different libraries from same individuals, using PMD and NO-PMD data, only-transversions genotypes vs standard genotypes and co-analysing new data with previously published data. |
| Randomization | No randomization was performed |
| Blinding | No blinding was performed. |

# Reporting for specific materials, systems and methods

We require information from authors about some types of materials, experimental systems and methods used in many studies. Here, indicate whether each material, system or method listed is relevant to your study. If you are not sure if a list item applies to your research, read the appropriate section before selecting a response.

## Materials & experimental systems

| n/a | Involved in the study |
|-----|----------------------|
| ☒ | ☐ Antibodies |
| ☒ | ☐ Eukaryotic cell lines |
| ☐ | ☒ Palaeontology and archaeology |
| ☒ | ☐ Animals and other organisms |
| ☒ | ☐ Clinical data |
| ☒ | ☐ Dual use research of concern |

## Methods

| n/a | Involved in the study |
|-----|----------------------|
| ☒ | ☐ ChIP-seq |
| ☒ | ☐ Flow cytometry |
| ☒ | ☐ MRI-based neuroimaging |

## Palaeontology and Archaeology

| | |
|---|---|
| Specimen provenance | Specimens come from excavations supported by Consejería de Cultura y Patrimonio Histórico de la Junta de Andalucía. |
| Specimen deposition | Specimens were available from Consejería de Cultura and Patrimonio Histórico de la Junta de Andalucía. |
| Dating methods | New AMS 14C dates were obtained from ultra-filtrated collagen. Collagen extraction was performed in the Department of Human Evolution at the Max Planck Institute for Evolutionary Anthropology (MPI-EVA), Leipzig, Germany and 14C measurements were done in Curt-Engelhorn-Zentrum Archäometrie gGmbH, Manheim. |

☒ Tick this box to confirm that the raw and calibrated dates are available in the paper or in Supplementary Information.

| | |
|---|---|
| Ethics oversight | The archaeological researchers supported by Consejería de Cultura y Patrimonio Histórico de la Junta de Andalucía approved and provided guidance on the study protocol. |

Note that full information on the approval of the study protocol must also be provided in the manuscript.

