## [Peer Review File · Nature Ecology & Evolution]

Peer Review Information

Journal: Nature Ecology & Evolution

Manuscript Title: The genetic link between Aurignacian and Magdalenian associated human groups in Ice Age western Europe

Corresponding author name(s): Vanessa Villalba-Mouco, Wolfgang Haak

Editorial Notes:

Reviewer Comments & Decisions:

Decision Letter, initial version:

2nd September 2022

Dear Vanessa

Your manuscript entitled "The genetic link between Aurignacian and Magdalenian associated human groups in Ice Age western Europe" has now been seen by three reviewers, whose comments are attached. The reviewers have raised a number of concerns which will need to be addressed before we can offer publication in Nature Ecology & Evolution. We will therefore need to see your responses to the criticisms raised and to some editorial concerns, along with a revised manuscript, before we can reach a final decision regarding publication.

Although there are extensive technical comments, reviewers are enthusiastic about the new genetic data, but from the archaeological reviewer's report it's clear that better integration with archaeological data is needed to substantiate the interpretations made--please pay particular attention to this report as you revise.

We therefore invite you to revise your manuscript taking into account all reviewer and editor comments. Please highlight all changes in the manuscript text file [OPTIONAL: in Microsoft Word format].

* If you have not done so already please begin to revise your manuscript so that it conforms to our Article format instructions at <http://www.nature.com/natecolevol/info/final-submission>. Refer also to any guidelines provided in this letter.

2* Include a revised version of any required reporting checklist. It will be available to referees (and, potentially, statisticians) to aid in their evaluation if the manuscript goes back for peer review. A revised checklist is essential for re-review of the paper.

[REDACTED]

Nature Ecology & Evolution is committed to improving transparency in authorship. As part of our efforts in this direction, we are now requesting that all authors identified as 'corresponding author' on published papers create and link their Open Researcher and Contributor Identifier (ORCID) with their account on the Manuscript Tracking System (MTS), prior to acceptance. ORCID helps the scientific community achieve unambiguous attribution of all scholarly contributions. You can create and link your ORCID from the home page of the MTS by clicking on 'Modify my Springer Nature account'. For more information please visit www.springernature.com/orcid.

[REDACTED]

Reviewer expertise:

Reviewer #1: signed report--Iberian Upper Palaeolithic archaeology

Reviewer #2: aDNA

Reviewer #3: aDNA

Reviewers' comments:

2Reviewer #1 (Remarks to the Author):

This paper deals with the genetic evidence of a Solutrean-associated individual from Cueva del Malalmuerzo and several samples from later chronological periods. This evidence serves to make the first arguments and hypotheses toward an understanding of the post-LGM occupation of the Iberian Peninsula and how it relates to the rest of Eurasia. The topic is fascinating and, in my opinion, relevant for Prehistory studies. Moreover, it is going to attract a lot of interest from geneticists and archaeologists. However, I believe the paper needs some more work (or maybe being a bit more careful) regarding offering a proper archaeological background to the research currently presented in the manuscript. I would like to stress that the only thing I want with my comments is to help the authors improve the text. I must say that I have learned a lot from reading the text and it has made me think about several issues regarding the Upper Palaeolithic.

Abstract

In the abstract there are acronyms such as "HG" and "LGM" I would avoid them. These acronyms are never introduced in the text either. Of course, most archaeologists will understand the meaning but I do not think is appropriate to include them in the abstract and not even clarify and introduce them later in the text.

Introduction section

The authors quickly cover the literature on the Solutrean in the Iberian Peninsula. However, the Gravettian and the Magdalenian are not properly introduced. For example, regarding the continuity between the Gravettian and Solutrean it was not Strauss the first one to propose this but Arrizabalaga (if I remember correctly). Moreover, the Gravettian has been well investigated in the last decade in the Iberian Peninsula with several Ph.D. thesis, syntheses, and articles. None of this literature is referenced, which is basic for the research introduction and the conclusions. The recent Gravettian research together with the previous works in the Atlantic area (Ph.D. of J. Zilhao, Almeida, etc.) in the Iberian Peninsula demonstrate that this technocomplex was not similar across the Peninsula but instead, there were very different techno traditions within it. This goes in accordance with other proposals in Western Europe that have defended a "mosaic" of techno traditions. I stress the case for the Gravettian but the same thing applies for the Aurignacian and Magdalenian, which are quite pivotal in the research but barely introduced in the manuscript.

The authors also make some statements that are speculative or they do not provide references for them, for example, they state:

- "In contrast to the preceding Gravettian and the subsequent Magdalenian when Northern Iberia was more densely occupied" What are the references for this? As far as I am concerned there is no robust evidence to sustain this.
- "Pre-LGM Gravettian-associated groups from central Europe differ genetically from post-LGM Magdalenian-associated groups, whereas the latter have been interpreted as being derived from a genetic ancestry first found in an Aurignacian-associated individual from northwestern Europe, and no western Gravettian individuals have been genetically analyzed so far". This information is essential. Is this data that the authors have produced? Is this statement base on previous publications? Again, references are lacking, and this information is crucial for the paper. Furthermore, the sentence is

3really long and a bit abstruse, as two ideas are presented. I would cut it in two and explain it better.

There is a lack of basic information to contextualize their analysis:

“Additionally, we sampled prehistoric human remains from southern Iberia from various cave and rock shelter sites (Supplementary 1) with long occupation histories with the ultimate aim to recover genome-wide ancient human DNA data from additional Upper Palaeolithic individuals, with a broader aim to establish a time transect in southern Iberia from the LGM to the Neolithic”.

Please provide the names of the sites and the industrial adscription and chronology in the main text. Moreover, it is not clear if this supplementary data is to enquire strictly on the Upper Palaeolithic or from the Upper Palaeolithic to the Neolithic. I think it is appropriate to clarify the aims since the beginning of the paper. What I want to say is that this paragraph is a bit ambiguous.

“ Fig. 1 | Chronological and geographical overview of newly reported and relevant published Individuals” which publications? References?

The genomic make-up of Solutrean hunther-gatherers of Cueva del Malalmuerzo section

“ These results suggest that the Solutrean-associated MLZ individual represents a lineage that is genetically intermediate between Goyet Q116-1 and Magdalenian-associated individuals, and that there was genetic continuity from the Solutrean to the Magdalenian period” the Solutrean and Magdalenian are not periods but technocomplex or technological traditions. I think is important that genetic papers are careful with the terminology used as they are implications for it.

In Fig. 2 I would clarify which technological tradition or technocomplex is associated with “Iberian HG” and “WHG”, Mesolithic?

“ Ultimately, this observation suggests a connection between Aurignacian and later Solutrean associated individuals in Western Europe, the IUP in central Europe and Tianyuan in the East, and that this genetic legacy persisted in Iberia for ~20,000 years more (MLZ, ~23 ka cal BP), while in Central (and presumably Eastern) Europe, it was superseded and already no longer traceable during the pre-Gravettian and Gravettian periods (~30 ka cal BP). The archaeological record provides evidence for the dispersal of the Aurignacian techno-complex in Iberia. Entering the peninsula from Southern France, archaeological remains securely assigned to the Proto- or Early Aurignacian are only found in Northern Iberia^{48–50}. For the sites Bajondillo near Málaga⁵¹ and Lapa do Picareiro in central Portugal⁵² the presence of an Early Aurignacian techno-complex has also been reported, but was challenged by several scholars^{50,53}. While sites with an Evolved Aurignacian techno-complex are attested to in the South, the presence of the Aurignacian is generally poorly represented.”

This genetic legacy could be associated with individuals with an Aurignacian technological tradition (in any of its variants) or even with other earlier technological traditions. With the evidence at hand, we only can hypothesize that. It must be bear in mind that “Initial Upper Palaeolithic” as far as I am

4concerned is not exactly the same as Aurignacian. Also, I am sure the authors are aware of this but the movement of people maybe does not always equal the movement of technotraditions.

In page 18 separate Moita Sebastiao.

Reading some parts of the text is quite difficult as the text is full of acronyms, abbreviations, and made-up synthesis names, even for a specialist (i.e. page 20). I am not saying this to criticize but for improving the manuscript. Also, next time provide the number of the lines in the manuscript, it will facilitate the reviewing process.

Conclusion section

“We find no substantial traces of admixture with the Gravettian-like ancestry in Southern Iberian, despite the spread of a common techno-complex horizon across Europe. Our results support a scenario in which the same techno-complex was shared by two distinct genetic ancestry groups, one in western Europe that is more similar to Goyet Q116-1 and at least one additional from central Europe that is genetically closer to Sunghir and Kostenki14”

The Gravettian has been stressed as a quite complex cultural tradition across Europe. The authors make this big archaeological statement without referencing any archaeological work. It is true that some authors have referred to the Gravettian as the first PanEuropean "culture" (e.g. Garrod, Otte, Simoneau), but many others have stressed how inside what we call now Gravettian there are many technological traditions. I would revise the archaeological literature for this. Do not get me wrong, I am not trying that the authors solve all the terminology problems related to the Upper Palaeolithic in Europe. However, when dealing with genetic information and archaeological terminology I would be careful in making direct and simplistic relationships. This can be solved, quite easily, by referencing the archaeological literature and being careful in some of the statements made. There are several archaeologists in the manuscript that surely will understand this call of attention.

Finally, I am a little bit surprised by the title “ The genetic link between Aurignacian and Magdalenian associated human groups in Ice Age western Europe” Their whole argument revolves around a purported individual associated with Solutrean technocomplex, I think I would stress that. Moreover, their paper has really interesting implications for the post-Palaeolithic period, they should maybe cover that in the title.

Supplementary material

First page: “contreras” the first letter should be capital, it is a surname

Paloma de la Peña

5Reviewer #2 (Remarks to the Author):

In the manuscript "The genetic link between Aurignacian and Magdalenian associated human groups in Ice Age western Europe", Villalba-Mouco et al. sequence genome-wide data from several ancient individuals in Iberia, dating from 23,000 years BP to Early Neolithic and Chalcolithic-related individuals. Focusing primarily on the 23,000 year BP individual (MLZ), they argue for an association with Magdalenian individuals but with excess early Asian-related ancestry similar to Q116-1. Overall, they use this finding to argue for two distinct European ancestries and one in Iberia that was present as far back as 23,000 years BP. They then explore the ancestral makeup of younger Iberian individuals sampled.

I do not think the manuscript is wrong on the point that it makes, but it is hard to see what is novel about their findings compared to that of previous studies they cited. The major issue is the lack of an introduction that sets up the previously known genetic findings, and the intermixing of results and discussions such that there is no clear set up as part of the final discussion showing what stands out from these results relative to the previous results that are referenced. The introduction is focused primarily on the greater archaeological context, but there have been important genetic findings such as the connection to Q116-1 of El Miron, the connection to Tianyuan broadly to many European individuals, and the Natufian connections more recently in the Villabruna cluster. Having the introduction set some of this up, and then indicate what is not known yet would then set up more clearly for why MLZ is especially pivotal for better understanding ancestral dynamics in Europe.

There are several conclusions that are arrived at using non-significant f_4 -statistics (ie, positive f_4 values where $|Z| < 3$). For example, p. 11 ($Z=2.011/2.244$), p. 14 ("However, the positive f_4 -statistic suggests a slightly higher amount of Villabruna-like ancestry in MLZ than in Goyet Q116-1"), and p. 16 ("All resulting f_4 -statistics were positive"). I would be careful making conclusions from these results, and use other significant results as your main point, following up with the observation of the positive trend. For the one on p. 11, would it make sense to try the same test using transversions as well? For p. 14, I would start with the $Z=2.972$ argument from the next sentence, and on p. 16, in Table 2.14, Z-scores were 2.058/1.917 for Natufian and Villabruna, but I could not find Taforalt results in the table.

For the younger individuals sequenced, the introduction to their data is rather abrupt. There is no discussion of data quality, familial relationships, and radiocarbon dating like there was for MLZ. Then, their general ancestry is not first discussed but is instead interspersed into later paragraphs. It seems useful to first orient the reader to whether they show similar ancestry to MLZ, vs Magdalenians, vs someone else (eg, other previously published Iberians?). On p. 18, it was argued that an individual from MoitaSebastiao (is that SE_Iberia_Meso in Table S2.15?) has high affinity to Tianyuan. However the number of SNPs available is 22,778 SNPs which seems low and makes the f_3 conclusion feel unreliable. Carihuela pops up multiple times in this and other similar analyses, but is not discussed throughout the text. I think this is a previously published individual, and if so, it is important to perhaps contextualize relative to this individual?

6The supplement seems to repeat some of the same text from the results/discussion, and introduce new f4-combinations (eg comparing to the Han) that is not mentioned at all in the main text. I usually expect everything in a supplemental note to be an expansion of details of a test or analysis that is summarized in the main text. I would work through the supplement and cut items that are already in the main manuscript or seem unnecessary to the main arguments, and add pieces that are important in the supplement to the right sections of the main text.

On p. 9, for f4(MLZ, Goyet Q116-1/Magdalenian; Věstonice-cluster, Mbuti), Fig S5B and S5C is cited, but should it be Figs S5A and S5B? Either way, I would suggest both figures be placed in Fig 2 instead of remaining in the supplement. On p. 11, "when Kostenki14 is chosen as a comparative UP baseline", Fig S5A is cited, but I think you mean Fig S5C.

On p. 11, "The Tianyuan-related ancestry present in MLZ is fully inherited from Goyet Q116-1 as both are cladal to Tianyuan individual using the f4-statistic of the form..." – Not sure how this conclusion is arrived at. All the f4-statistic shows is that there is a similar relationship of MLZ and Q116-1 to Tianyuan, but that could also be similar levels from two separate contributions rather than inheritance from Q116-1. F4-statistics don't typically show cladal relationships using a single f4-combination.

Reviewer #3 (Remarks to the Author):

The manuscript has no major flaws, although the data is limited, and additional data would be needed to allow a robust interpretation of the genetic profile of the region and cultural complex.

Conclusions are original, as this is the first genome from the Iberian Peninsula dated to the LGM (alongside a number of more recent genomes that assist the contextualisation of the unique ancestry pattern).

The sequence data will be incorporated into future studies and analyses by those within the field. The analysis of the MLZ genome suggest populations that as yet haven't been identified through genome sequencing, and thus adds further depth to our understanding of the human populations of the LGM I would recommend this for publication. It presents and contextualises the first genomic data from the Iberian Peninsula dated to the peak of the last glacial period, a period from which the potential for human sequence data is sparse.

The "Solutrean" MLZ genome shows a previously unseen genetic ancestry derived from a population ancestral to those inhabiting the LGM refugia in Italy, and a population with an affinity to the Goyet116 individual in Belgium and dated to 35kya (Aurigean pop). Further, a degree of long-term genetic continuity is suggested, with the MLZ genome showing affinity to an unsequenced but predicted population ancestral to some (but not all) of the oldest European human genomes.

The results shows a degree of genetic continuity in the region during the latter half of the LGM, suggesting a stable and long-established human population with limited inward migration and apparently no widescale genetic turnover

No novel methods are used, although all are up-to-date. The use of both full and PMDtools subest sequence files are a very nice method to add support to results.

Additional notes:

7Introduction:

"Human presence in the archaeological record is documented predominantly by artifacts, mainly stone tools assigned to so-called techno-complexes, rather than by skeletal remains"

I am not sure this is the case for all complexes mentioned: The Magdalenian is characterised by cave art, tools made out of faunal remains and cut marked human remains; the Gravettian by Venus figurines (how are you defining artefacts here?)

"Pre-LGM Gravettian-associated groups from central Europe differ genetically from post-LGM Magdalenian-associated groups, whereas the latter have been interpreted as being derived from a genetic ancestry first found in an Aurignacian-associated individual from north-western Europe, and no western Gravettian individuals have been genetically analysed so far. As a consequence, the genomic processes accompanying the cultural transition from the Gravettian to the Magdalenian across Europe remain poorly understood, especially in Western Europe" -
This seems unreferenced, (Fu 2016 reference ?)

Figure 1b and c:

Why is El Miron classed as an Iberian HS/Epipalaeolithic/Meso hunter-gatherer despite the age of the specimen?

Results:

Lines 5-10

Breaking the 8 individuals down into origin (3 Solutrean, 5 more recent) would make it easier to follow.

"the fact that we could not select skeletal elements that were shown to yield a high amount of endogenous human DNA"

Change to "...have been shown...". And why? Perhaps the material was too fragmented? No curatorial permission?

"Testing for biological relatedness between the two individuals using the pairwise-mismatch rate (PMR) revealed unusually low values of 0.15246 (Supplementary 2, Table S1.5). After excluding four libraries from MLZ003 with signs of contamination, the PMR value was found to be even lower (0.1039)"

I would like to be much more confident that there is a sound statistical basis for combining the data from these two elements. Alternatively, is there potential to run the analyses with each separated to determine whether results are congruent, as one would expect from the same individual. Please also include more information on the two teeth.

"The final coverage of MLZ003005 was 0.41X, corresponding to 226,914 autosomal SNPs in the 1240k panel"

I'd like to understand how this coverage was determined. Given the data is SNP capture, this 0.41x coverage isn't likely to be whole genome coverage. If so, how has this been calculated? Is this instead average coverage of all 1240k sites?

Figure 4: "Results show that MLZ is the only Pleistocene HG that indicated a significant and equal

8attraction to both Natufian and Villabruna-like ancestries "

What about Paglicci133 individual in same figure? This would be easier to follow if Pleistocene HGs were defined in the text/figure.

Great use of the PMD tools subsetted bam alongside the whole bam. However, could be clearer in the main text as to the extent of this subset (prop. of original/whole bam). See supplemental comments.

Conclusions:

"Genome-wide data from the first Solutrean-associated individual MLZ in southern Iberia revealed a genetic link with an IUP population, which also connects IUP individuals Bacho Kiro, Tianyuan and Goyet Q116-1 from regions in today's Belgium, Bulgaria and China. The same ancestral population later split into two groups, which contributed to European and Asian populations, respectively" Perhaps change "revealed a genetic link" to "is descended from". Clarify what is meant by "also connects".

Material and Methods:

Methods contemporary. As suggested, please add additional analysis to add certainty to sample combination.

"Libraries with >0.1 % endogenous DNA, were subsequently enriched for ~1240k single nucleotide polymorphisms (SNPs)... "

In the results section the value was 0.08%

Supplementals:

Capitalise "Francisco contreras" (?)

What type/form of tooth were sample ie (canines/molars etc)? Would also provide info to further support the combination of the two samples/

2.2 PMD filtering section

Supplemental information incorrect: please check this.

Loss read % indicates the % reads retained, not removed. Therefore, ADS001 actually the best-preserved sample. Mix up in the % and sample name in the text here.

PMD score of below 3 doesn't directly correspond with contamination, but rather the degree of deamination present which is a predictor of contamination. Is contamination the correct word at all?

Supplemental figure S1c

There is a slight error here ; PDM >>> PMD

*****END*****

Author Rebuttal to Initial comments

Reviewer expertise:

Reviewer #1: signed report--Iberian Upper Palaeolithic archaeology

Reviewer #2: aDNA

Reviewer #3: aDNA

Reviewers' comments:

We would like to thank all three reviewers for the positive assessment and constructive feedback that has helped to improve our manuscript considerably. Please find our responses in blue Italics below.

Reviewer #1 (Remarks to the Author):

This paper deals with the genetic evidence of a Solutrean-associated individual from Cueva del Malalmuerzo and several samples from later chronological periods. This evidence serves to make the first arguments and hypotheses toward an understanding of the post-LGM occupation of the Iberian Peninsula and how it relates to the rest of Eurasia. The topic is fascinating and, in my opinion, relevant for Prehistory studies. Moreover, it is going to attract a lot of interest from geneticists and archaeologists. However, I believe the paper needs some more work (or maybe being a bit more careful) regarding offering a proper archaeological background to the research currently presented in the manuscript. I would like to stress that the only thing I want with my comments is to help the authors improve the text. I must say that I have learned a lot from reading the text and it has made me think about several issues regarding the Upper Palaeolithic.

We appreciate your comments. We think that the manuscript has now been substantially improved, becoming more technically precise with respect to the archaeological aspects, but also more digestible for non-aDNA experts.

We have described the current state of the knowledge in both of the fields of archaeology and archaeogenetics, describing what is known from the time period in the region of interest, and provide a wider overview of the genetic landscape in Europe (including more detailed information about Western Europe and Iberia specifically). We have tried to integrate the description of genetics and archaeology as much as possible, keeping in mind that changes in the archaeological record do not always coincide with changes in the genomic makeup of the hunter-gatherer populations. We have now stressed this important point in the introduction, too.

Abstract

In the abstract there are acronyms such as "HG" and "LGM" I would avoid them. These acronyms are never introduced in the text either. Of course, most archaeologists will understand the meaning but I do not think is appropriate to include them in the abstract and not even clarify and introduce them later in the text.

We have taken care to add the definition of each acronym in every instance of their first use, both in the abstract and the main manuscript separately.

Now that we have extended the genetics introduction, the term “Hunter-gatherer (HG)” appears relatively often and would stretch the number of characters substantially. The term is broadly used in the genetic literature for many Epipaleolithic and Mesolithic populations, which are genetically differentiated and thus require a distinctive nomenclature (see e.g. Western Hunter-gatherers, [WHG]; Iberian Hunter-gatherers, [Iberian HG]).

Introduction section

The authors quickly cover the literature on the Solutrean in the Iberian Peninsula. However, the Gravettian and the Magdalenian are not properly introduced. For example, regarding the continuity between the Gravettian and Solutrean it was not Strauss the first one to propose this but Arrizabalaga (if I remember correctly). Moreover, the Gravettian has been well investigated in the last decade in the Iberian Peninsula with several Ph.D. theses, syntheses, and articles. None of this literature is referenced, which is basic for the research introduction and the conclusions. The recent Gravettian research together with the previous works in the Atlantic area (Ph.D. of J. Zilhao, Almeida, etc.) in the Iberian Peninsula demonstrate that this technocomplex was not similar across the Peninsula but instead, there were very different techno traditions within it. This goes in accordance with other proposals in Western Europe that have defended a “mosaic” of techno traditions. I

stress the case for the Gravettian but the same thing applies for the Aurignacian and Magdalenian, which are quite pivotal in the research but barely introduced in the manuscript.

We have now extended the Introduction about the Solutrean technocomplex, which provides the technological, and, to an extent, chronological (but see below), context of the data presented in this study. We thank the reviewer for the corrections and suggestions in the bibliography regarding the Magdalenian and the Gravettian. We have used these in the discussion and in the conclusions about implications of the Gravettian and Magdalenian, and have adjusted the introduction accordingly. Here, we now state that “Gravettian-associated individuals from pre-LGM central Europe and Italian Peninsula form a genetic cluster (i.e. share more ancestry within the group than with individuals outside the group) (Fig. 1), which was named after the oldest individual, here Véstonice cluster (Fu et al. 2016), irrespective of the various Gravettian industries they were associated with archaeologically.”. Unfortunately we do not have the genetic resolution that would be required to differentiate between individuals who were assigned to various central European assemblage/techno-complex traditions.

We cited Strauss 2000, in which we found the following passage, albeit without citation: “The technological traditions of the Franco-Iberian Solutrean were firmly rooted in those of the Gravettian (middle Upper Paleolithic) of western Europe.”

We thus assumed that it was Strauss who had first made this statement. We thank the reviewer for pointing out our mistake. We have followed your suggestions and added earlier references from Zilhão and Aubry (1995), Zilhão et al 1999 and Almeida (2006), who - as we have learned - had suggested earlier that the Solutrean were a continuation of the Gravettian traditions, more so than a complete breakdown. We also added references which are in line with this hypothesis in the region of today’s France (Renard et al. 2012), and other references, which explore the continuity from Gravettian to Solutrean in other disciplines, such as the study of mobile art (e.g. Villalverde et al. 2009).

We hope to have achieved a better overview in both of the fields of genomics and archaeology.

The authors also make some statements that are speculative or they do not provide references for them, for example, they state:

- "In contrast to the preceding Gravettian and the subsequent Magdalenian when Northern Iberia was more densely occupied" What are the references for this? As far as I am concerned there is no robust evidence to sustain this.

In order to support this statement, we had provided two references in line 122 in the original submission: Schmidt et al. 2012 and Weniger et al. 2019. Schmidt et al. 2012 show a map with the distribution of Solutrean sites in the Northern and Southern Iberian regions, which reflects the difference in occupation density. Additionally, Schmidt et al. 2012 conclude that "Although for reasons presently unclear, an entirely new settlement scenario can be observed for the Solutrean. For the first time in the Late Pleistocene of Iberia, the presence of humans in the landscape is now similarly well-documented in the south and in the north. This is a major difference with previous periods".

Weniger et al. 2019 analyzed the Magdalenian settlement of Iberia using the same methods as Schmidt et al. 2012, reporting a decrease in human presence in Southern Iberia at that time.

- "Pre-LGM Gravettian-associated groups from central Europe differ genetically from post-LGM Magdalenian-associated groups, whereas the latter have been interpreted as being derived from a genetic ancestry first found in an Aurignacian-associated individual from northwestern Europe, and no western Gravettian individuals have been genetically analyzed so far". This information is essential. Is this data that the authors have produced? Is this statement based on previous publications? Again, references are lacking, and this information is crucial for the paper. Furthermore, the sentence is really long and a bit abstruse, as two ideas are presented. I would cut it in two and explain it better.

The statement is based on the findings published in Fu et al. 2016, a landmark genomic reference dataset of Paleolithic and Mesolithic Hunter-gatherers. We have made sure that the citation appears in the correct position now. We thank the reviewer for pointing this out.

There is a lack of basic information to contextualize their analysis:

"Additionally, we sampled prehistoric human remains from southern Iberia from various cave and rock shelter sites (Supplementary 1) with long occupation histories with the ultimate aim to recover genome-wide ancient human DNA data from additional Upper Paleolithic individuals, with a broader aim to establish a time transect in southern Iberia from the LGM to the Neolithic".

Please provide the names of the sites and the industrial adscription and chronology in the main text. Moreover, it is not clear if this supplementary data is to enquire strictly on the Upper Palaeolithic or from the Upper Palaeolithic to the Neolithic. I think it is appropriate to clarify the aims since the beginning of the paper. What I want to say is that this paragraph is a bit ambiguous.

A more detailed description of the other individuals included in this study has been added.

" Fig. 1 | Chronological and geographical overview of newly reported and relevant published Individuals" which publications? References?

We have compiled all of this information in Table S1.1 that now is referred to in Fig. 1, in order to avoid overcrowding of Fig. 1. Table S1.1 lists the archaeological codes, the genetic labels, chronology, location and genetic data such as the number of autosomal SNPs reported, and uniparentally-inherited markers (MT and Y haplogroups).

The genomic make-up of Solutrean hunter-gatherers of Cueva del Malalmuerzo section

" These results suggest that the Solutrean-associated MLZ individual represents a lineage that is genetically intermediate between Goyet Q116-1 and Magdalenian-associated individuals, and that there was genetic continuity from the Solutrean to the Magdalenian period" the Solutrean and Magdalenian are not periods but technocomplex or technological traditions. I think is important that genetic papers are careful with the terminology used as they are implications for it.

We agree in parts and have now changed this statement to read as follows: " These results suggest that the Solutrean-associated MLZ individual represents a lineage that is genetically intermediate between Goyet Q116-1 and individuals from the Goyet Q2 cluster, following a chronological argument where Goyet Q116-1 is more genetically similar to MLZ than to the Goyet Q2 cluster, but where MLZ is also genetically more similar to the Goyet Q2 cluster than to the preceding Goyet Q116-1 (Fig. 2b and 2c). Identifying MLZ as the ancestral lineage of the Goyet Q2 cluster (i.e., Magdalenian-associated individuals) is in agreement with the archeological record that postulates the emergence of the Magdalenian techno-complex in regions of northern Iberia and southern France^{44,45}."

We are aware that techno-complexes cannot be equated with chronological periods, but since these technological traditions follow a chronological order within the respective time period of the Upper Paleolithic, a certain periodicity cannot be denied, in particular since these traditions are dated as well. The same principle applies to later archaeological cultures or phenomena that have defined a chronological era and come with relatively clearly defined age ranges. However, we have taken care to not equate and conflate these terms nonchalantly wherever possible. In addition, when applicable, we prefer not to talk about "ancestry profiles" of techno-complexes (indicating population homogeneity within), but instead discuss the ancestry profiles of securely dated individuals who are associated with these techno-complexes.

In Fig. 2 I would clarify which technological tradition or technocomplex is associated with "Iberian HG" and "WHG", Mesolithic?

We have now added the explanation in the introduction chapter.

" Ultimately, this observation suggests a connection between Aurignacian and later Solutrean associated individuals in Western Europe, the IUP in central Europe and Tianyuan in the East, and that this genetic legacy persisted in Iberia for ~20,000 years more (MLZ, ~23 ka cal BP), while in Central (and presumably Eastern) Europe, it was superseded and already no longer

traceable during the pre-Gravettian and Gravettian periods (~30 ka cal BP). The archaeological record provides evidence for the dispersal of the Aurignacian techno-complex in Iberia. Entering the peninsula from Southern France, archaeological remains securely assigned to the Proto- or Early Aurignacian are only found in Northern Iberia^{48–50}. For the sites Bajondillo near Málaga⁵¹ and Lapa do Picareiro in central Portugal⁵² the presence of an Early Aurignacian techno-complex has also been reported, but was challenged by several scholars^{50,53}. While sites with an Evolved Aurignacian techno-complex are attested to in the South, the presence of the Aurignacian is generally poorly represented.”

This genetic legacy could be associated with individuals with an Aurignacian technological tradition (in any of its variants) or even with other earlier technological traditions. With the evidence at hand, we only can hypothesize that. It must be bear in mind that “Initial Upper Palaeolithic” as far as I am concerned is not exactly the same as Aurignacian. Also, I am sure the authors are aware of this but the movement of people maybe does not always equal the movement of technotraditions.

Thank you. We are aware that Initial Upper Paleolithic technologies are not the same as the Aurignacian. However, from a genetic perspective, the Aurignacian-associated individual Goyet Q116-1 still carries ancestry from some IUP-associated individuals, such as Bacho Kiro and Tianyuan. Other individuals associated with IUP techno-traditions seem to not have contributed to later populations, with their ancestry going completely extinct (e.g. Zlatý kůň).

However, the ancestry carried by MLZ, derived from individuals as temporally distal as the IUP, is also found in the Aurignacian-associated individual Goyet Q116-1. As a consequence, we now suggest that the ancestry found during the IUP was brought to Southern Iberia by individuals associated with the Aurignacian (sensu lato) as the archeological record only provides evidence of IUP industries (e.g. Chatelperronian) in Northern Iberia.

In page 18 separate Moita Sebastiao.

We agree. The site name Moita do Sebastião is now written out and reported properly.

Reading some parts of the text is quite difficult as the text is full of acronyms, abbreviations, and made-up synthesis names, even for a specialist (i.e. page 20). I am not saying this to criticize but for improving the manuscript. Also, next time provide the number of the lines in the manuscript, it will facilitate the reviewing process.

Acronyms such as CA (Copper Age) and EN (Early Neolithic) are now introduced in the Introduction chapter. We have double-checked that all acronyms were introduced and defined correctly.

We also provide line numbers in the revised version of the manuscript. Apologies for the omission during the first submission.

Conclusion section

“We find no substantial traces of admixture with the Gravettian-like ancestry in Southern Iberian, despite the spread of a common

techno-complex horizon across Europe. Our results support a scenario in which the same techno-complex was shared by two distinct genetic ancestry groups, one in western Europe that is more similar to Goyet Q116-1 and at least one additional from central Europe that is genetically closer to Sungir and Kostenki14”

The Gravettian has been stressed as a quite complex cultural tradition across Europe. The authors make this big archaeological statement without referencing any archaeological work. It is true that some authors have referred to the Gravettian as the first PanEuropean "culture" (e.g. Garrod, Otte, Simoneau), but many others have stressed how inside what we call now Gravettian there are many technological traditions. I would revise the archaeological literature for this. Do not get me wrong, I am not trying that the authors solve all the terminology problems related to the Upper Palaeolithic in Europe. However, when dealing with genetic information and archaeological terminology I would be careful in making direct and simplistic relationships. This can be solved, quite easily, by referencing the archaeological literature and being careful in some of the statements made. There are several archaeologists in the manuscript that surely will understand this call of attention.

*Thank you. It was not our intention to make an archaeological statement using genetic data. In fact, we are aware that our genomic analyses do not always have the same regional resolution as defined by the lithic technologies. Our aim was to indirectly suggest that different genetic ancestries can be associated with the Gravettian techno-complex sensu lato. This is a novel insight as analyses of human genomic data from Gravettian-associated individuals have so far shown strong genetic within-group similarities, and as a result, all individuals had been assigned to the Věstonice cluster (Fu et al. 2016). Our new data now support a different scenario, despite lacking direct Gravettian associated individuals from western Europe, and this scenario also contradicts the idea of a "Pan-European culture" as suggested by e.g. Otte (1981). We have now clarified this, and cite the references provided above in the **Results and Discussion** section, now reading as follows:*

"This means that the genetic discontinuity between pre-LGM and post-LGM, as reported by (Fu et al. 2016), was not driven by the harsh climatic change, as it is already observed in MLZ, an individual directly dated to the height of the LGM. These results suggest that the apparent genetic discontinuity might be better explained by geography, which would imply that during the time when the Gravettian techno-complex prevailed, at least two genetically distinct groups/populations must have existed in Europe: one in western Europe, which was more genetically similar to Goyet Q116-1, and a second in central (and perhaps eastern) Europe, which had been described as the Věstonice cluster³⁷. Our results are in line with technological studies which suggest that the Solutrean technocomplex was rooted in Western Gravettian technologies (Zilhão and Aubry 1995; Zilhão et al. 1999; Renard 2011), and that the continuity/resemblance of the rock art associated with the Gravettian and Solutrean traditions in Western Europe (Villaverde et al. 2009). In contrast, this result would rule out the monocentric central European origin of the Gravettian technocomplex (Otte 1981)."

*We have added all references in the Results and Discussion to avoid a massive number of citations in the **Conclusions**.*

Finally, I am a little bit surprised by the title "The genetic link between Aurignacian and Magdalenian associated human groups in Ice Age western Europe" Their whole argument

revolves around a purported individual associated with Solutrean technocomplex, I think I would stress that. Moreover, their paper has really interesting implications for the post-Palaeolithic period, they should maybe cover that in the title.

We have changed the title, now reading: "A 23,000-year-old southern-Iberian individual links human groups that lived in Western Europe before and after the Last Glacial Maximum", putting emphasis on the date of the key individual and its genetic legacy on post-LGM populations.

Supplementary material

First page: "contreras" the first letter should be capital, it is a surname

Corrected, thank you!

Paloma de la Peña

Reviewer #2 (Remarks to the Author):

In the manuscript "The genetic link between Aurignacian and Magdalenian associated human groups in Ice Age western Europe", Villalba-Mouco et al. Sequence genome-wide data from several ancient individuals in Iberia, dating from 23,000 years BP to Early Neolithic and Chalcolithic-related individuals. Focusing primarily on the 23,000 year BP individual (MLZ), they argue for an association with Magdalenian individuals but with excess early Asian-related ancestry similar to Q116-1. Overall, they use this finding to argue for two distinct European ancestries and one in Iberia that was present as far back as 23,000 years BP. They then explore the ancestral makeup of younger Iberian individuals sampled.

I do not think the manuscript is wrong on the point that it makes, but it is hard to see what is novel about their findings compared to that of previous studies they cited. The major issue is the lack of an introduction that sets up the previously known genetic findings, and the intermixing of results and discussions such that there is no clear set up as part of the final discussion showing what stands out from these results relative to the previous results that are referenced. The introduction is focused primarily on the greater archaeological context, but there have been important genetic findings such as the connection to Q116-1 of El Miron, the connection to Tianyuan broadly to many European individuals, and the Natufian connections more recently in the Villabruna cluster. Having the introduction set some of this up, and then indicate what is not known yet would then set up more clearly for why MLZ is especially pivotal for better understanding ancestral dynamics in Europe.

Thank you. We completely agree with the fact that the current state of knowledge in the field of archaeogenetics relating to these time periods was not discussed in the introduction in sufficient detail. We have now added more information about what is known and explain why MLZ is a key individual for understanding the population dynamics in western Europe.

We think that combining the results and their explanation/discussion would help the general reader, but agree that this was not always done well, or was not well supported/complemented

by the bibliography from both of the fields genetics and archaeology. We have now complemented the results with better explanations, making sure that all of the required citations have been added.

There are several conclusions that are arrived at using non-significant f_4 -statistics (ie, positive f_4 values where $|Z| < 3$). For example, p. 11 ($Z = 2.011/2.244$), p. 14 ("However, the positive f_4 -statistic suggests a slightly higher amount of Villabruna-like ancestry in MLZ than in Goyet Q116-1"), and p. 16 ("All resulting f_4 -statistics were positive"). I would be careful making conclusions from these results, and use other significant results as your main point, following up with the observation of the positive trend. For the one on p. 11, would it make sense to try the same test using transversions as well? For p. 14, I would start with the $Z = 2.972$ argument from the next sentence, and on p. 16, in Table 2.14, Z-scores were 2.058/1.917 for Natufian and Villabruna, but I could not find Taforal results in the table.

Many thanks for these suggestions. Following these recommendations, we now start the paragraph with the most significant result obtained from $f_4(MLZ, Kostenki14; Villabruna, Mbuti)$, Z -score = 2.972 and then observations that point to the same trend. We also point out that while we use a Z-score cut off of $|Z| < 3$, that the true theoretical distribution of f_3 - and f_4 -statistics is not known, making this a conservative cut off. Hence, there is still some potential value in discussing f_3 - and f_4 -statistics that are close to this cut off, and additional value in trends of positive (and hence not randomly distributed around zero) f -statistics.

Taforal was labeled as "Morocco_Iberomaurusian". We have now made sure that the population is labeled consistently throughout the manuscript. Thank you for pointing this out.

We have genotyped MLZ for transversions exclusively (MLZ005003.TV) and have re-run all analyses with the three versions of data: MLZ005003 (standard 1240k pulldown version), MLZ005003.PMD (PMD-filtered version) and MLZ005003.TV (transversions only). MLZ005003.TV accounts for 65,431 SNPs, which drastically decreases the power of resolution, resulting in wider error bars. However, the results provided by MLZ005003.TV still confirm the original result. We present these new results in all supplementary tables, and have included MLZ005003.TV in the three PCA plots of Figure S3.

For the younger individuals sequenced, the introduction to their data is rather abrupt. There is no discussion of data quality, familial relationships, and radiocarbon dating like there was for MLZ. Then, their general ancestry is not first discussed but is instead interspersed into later paragraphs. It seems useful to first orient the reader to whether they show similar ancestry to MLZ, vs Magdalenians, vs someone else (eg, other previously published Iberians?). On p. 18, it was argued that an individual from MoitaSebastiao (is that SE_Iberia_Meso in Table S2.15?) has high affinity to Tianyuan. However the number of SNPs available is 22,778 SNPs which seems low and makes the f_3 conclusion feel unreliable. Carihuela pops up multiple times in this and other similar analyses, but is not discussed throughout the text. I think this is a previously published individual, and if so, it is important to perhaps contextualize relative to this individual?

Thanks. We agree that the introduction was very abrupt and largely biased towards the oldest individual from Malalmuerzo. We have now included more information about the chronologically younger individuals from Andalucia and the rationale for co-analysing them,

and clarified the hypotheses that we wanted to test. Due to the limitations of space, we have kept the quality filtering steps and the test for biological relatedness in the supplement, but added cross-references in the main text.

The label *SE_Iberia_Meso* is the combination of two low-coverage HGs from Cingle del Mas Nou and Cueva de la Cocina published in Olalde et al. 2019. The individual from Moita do Sebastião has in fact >200,000 SNPs. La Carihuela cannot be discussed in detail due to the lack of a direct radiocarbon date. This individual was published in Olalde et al. 2019, and the authors specifically mentioned that direct dating had failed. As for all of the Iberian HGs, the Carihuela individual retained high levels of ancestry from the preceding Goyet Q2 cluster, but it is impossible to differentiate whether La Carihuela comes from a Magdalenian or a Mesolithic context as in Southern Iberia all Mesolithic individuals are also highly enriched in Goyet Q2 ancestry. We have now added a Supplementary Table (Table S1.1) which lists all of the analyzed HGs, and the relevant information such as coverage, chronology and publication in order to avoid misunderstandings.

The supplement seems to repeat some of the same text from the results/discussion, and introduce new f_4 -combinations (eg comparing to the Han) that is not mentioned at all in the main text. I usually expect everything in a supplemental note to be an expansion of details of a test or analysis that is summarized in the main text. I would work through the supplement and cut items that are already in the main manuscript or seem unnecessary to the main arguments, and add pieces that are important in the supplement to the right sections of the main text.

Thanks. We agree and have shortened some parts of the Supplementary information that did not provide additional information for what is presented in the main manuscript. Specifically, we have removed information from the subheader “4. Relationship of MLZ to other ancient individuals” that was duplicated in the manuscript.

Regarding the f_4 -outgroup test using Han, we think it is important to keep it in the current context as a comparison with another f_4 -statistic that tests for Basal Eurasian ancestry, which is $f_4(\text{Kostenki14, Ust-Ishim; test, Mbuti})$. Both tests, the one published by Yang et al. 2018 and ours, follow the same rationale and yield consistent results. However, figure S6 presents only our proposed test $f_4(\text{Kostenki14, Ust-Ishim; test, Mbuti})$ but in Table S2.12, both tests $f_4(\text{Kostenki14, Ust-Ishim; test, Mbuti})$ and $f_4(\text{Han, Ust-Ishim; test, Mbuti})$ are shown. On page 20, we mentioned that several tests have been performed to exclude Basal Eurasian ancestry in the MLZ individual, which reads as follows: “By contrast, we found no indication for Basal Eurasian ancestry in MLZ using other tests”.

On p. 9, for $f_4(\text{MLZ, Goyet Q116-1/Magdalenian; Věstonice-cluster, Mbuti})$, Fig S5B and S5C is cited, but should it be Figs S5A and S5B? Either way, I would suggest both figures be placed in Fig 2 instead of remaining in the supplement. On p. 11, “when Kostenki14 is chosen as a comparative UP baseline”, Fig S5A is cited, but I think you mean Fig S5C.

Thanks. The figures that showed the results of the f_4 -statistics of the form $f_4(\text{MLZ, Goyet Q116-1/Goyet Q2 cluster; Věstonice-cluster, Mbuti})$, previously Fig. S5a and S5a, are now part of panel 2 as per the reviewer's suggestion.

All panels in this figure have been renumbered accordingly.

On p. 11, "The Tianyuan-related ancestry present in MLZ is fully inherited from Goyet Q116-1 as both are cladal to Tianyuan individual using the f_4 -statistic of the form..." – Not sure how this conclusion is arrived at. All the f_4 -statistic shows is that there is a similar relationship of MLZ and Q116-1 to Tianyuan, but that could also be similar levels from two separate contributions rather than inheritance from Q116-1. F_4 -statistics don't typically show cladal relationships using a single f_4 -combination.

Using f_4 -statistics of the form $f_4(\text{MLZ}, \text{Goyet Q2 cluster}; \text{Goyet Q116-1}, \text{Mbuti})$ (Figure 2c) we first observed that there was a close genetic relationship between MLZ and Goyet Q116-1 when compared with Magdalenian-associated individuals such as El Mirón, Goyet Q2 or Hohle Fels 49. The result of this test provided evidence that MLZ shares an excess of genetic ancestry with Goyet Q116-1 that is not shared by other individuals from the Goyet Q2 cluster.

Shared genetic drift between MLZ and Goyet Q116-1 is also seen in the result of the f_4 -statistic of the form $f_4(\text{MLZ}, \text{Kostenki 14}; \text{test}, \text{Mbuti})$, when 'test' is Goyet Q116-1 (shown in Figure 3a). This result is linked to the one presented in Figure 3b where we reported similar levels of shared genetic drift between Tianyuan and Goyet Q116-1 and Tianyuan and MLZ.

Moreover, the similarity of Goyet Q116-1 and MLZ becomes clear when we test for excess shared drift with any Initial Upper Paleolithic (IUP) individual, e.g. Bacho Kiro IUP and Tianyuan. The results for the f_4 -statistics of the form $f_4(\text{Goyet Q116-1}, \text{MLZ}; \text{Bacho Kiro IUP/Tianyuan}, \text{Mbuti})$ are all consistent with zero and are presented now in Table S2.7. All f_4 -statistics in which MLZ and Goyet Q116-1 were set as comparisons with IUP individuals resulted in symmetrical relationships. However, f_4 -statistics of the form $f_4(\text{Goyet Q116-1}, \text{MLZ}; \text{test}, \text{Mbuti})$ indicate an asymmetrical relationship between Goyet Q116-1 and MLZ when testing WHG-, or Goyet Q2-like individuals who carry excess shared drift with MLZ. This suggests that the excess of shared affinity must be the result of later admixture events, which were present in MLZ but not in Goyet Q116-1 (Table S2.14).

Finally, our qpGraph model also statistically supports the hypothesis that MLZ inherited Tianyuan-related ancestry from Goyet Q116-1-like ancestry, rather than receiving it through an independent pulse. The most parsimonious explanation for the affinity of MLZ to Tianyuan and Bacho Kiro IUP is that MLZ is derived from a Goyet Q116-1-like population. Archaeologically, there is no evidence of any IUP technology in Southern Iberia that could have brought this genetic ancestry independently.

We have now added the new results to table S2.7, discussed them in the main manuscript in support of our arguments, and have updated the respective paragraphs in the main manuscript accordingly.

Reviewer #3 (Remarks to the Author):

The manuscript has no major flaws, although the data is limited, and additional data would be needed to allow a robust interpretation of the genetic profile of the region and cultural complex.

Conclusions are original, as this is the first genome from the Iberian Peninsula dated to the LGM (alongside a number of more recent genomes that assist the contextualisation of the unique ancestry pattern).

The sequence data will be incorporated into future studies and analyses by those within the field. The analysis of the MLZ genome suggest populations that as yet haven't been identified through genome sequencing, and thus adds further depth to our understanding of the human populations of the LGM

I would recommend this for publication. It presents and contextualises the first genomic data from the Iberian Peninsula dated to the peak of the last glacial period, a period from which the potential for human sequence data is sparse.

The "Solutrean" MLZ genome shows a previously unseen genetic ancestry derived from a population ancestral to those inhabiting the LGM refugia in Italy, and a population with an affinity to the Goyet116 individual in Belgium and dated to 35kya (Aurigeon pop). Further, a degree of long-term genetic continuity is suggested, with the MLZ genome showing affinity to an unsequenced but predicted population ancestral to some (but not all) of the oldest European human genomes.

The results shows a degree of genetic continuity in the region during the latter half of the LGM, suggesting a stable and long-established human population with limited inward migration and apparently no widescale genetic turnover

No novel methods are used, although all are up-to-date. The use of both full and PMDtools subest sequence files are a very nice method to add support to results.

Many thanks. We appreciate the positive feedback.

Additional notes:

Introduction:

"Human presence in the archaeological record is documented predominantly by artifacts, mainly stone tools assigned to so-called techno-complexes, rather than by skeletal remains" I am not sure this is the case for all complexes mentioned: The Magdalenian is characterised by cave art, tools made out of faunal remains and cut marked human remains; the Gravettian by Venus figurines (how are you defining artefacts here?)

Thanks. This is very much in line with suggestions from Reviewer #1, upon which we have expanded and improved the archaeological background in the introduction and discussion sections.

"Pre-LGM Gravettian-associated groups from central Europe differ genetically from post-LGM Magdalenian-associated groups, whereas the latter have been interpreted as being derived from a genetic ancestry first found in an Aurignacian-associated individual from north-western Europe, and no western Gravettian individuals have been genetically analysed so far. As a consequence, the genomic processes accompanying the cultural transition from the Gravettian to the Magdalenian across Europe remain poorly understood, especially in Western Europe" - This seems unreferenceed, (Fu 2016 reference ?)

Thanks. Yes, this reference was missing and has now been added. We have also updated the introduction with a summary about the genetic history of hunter-gatherers, with a main focus in Western Europe. We hope the introduction reads more clearly now.

Figure 1b and c:

Why is El Miron classed as an Iberian HS/Epipalaeolithic/Meso hunter-gatherer despite the age of the specimen?

An explanation for this classification is now given in the revised introduction section. In brief, Iberian hunter-gatherer individuals are a good example of why genetic clusters and “cultural” traditions do not match perfectly.

Results:

Lines 5-10

Breaking the 8 individuals down into origin (3 Solutrean, 5 more recent) would make it easier to follow.

Thanks. This has been done.

“the fact that we could not select skeletal elements that were shown to yield a high amount of endogenous human DNA”

Change to “...have been shown...”. And why? Perhaps the material was too fragmented? No curatorial permission?

Here we refer specifically to the petrous bone that was absent from the skeletal material we were able to take samples from. This is now explicitly mentioned. Of note, as pointed out in the archaeological descriptions, most find contexts consist of scattered stray finds from various strata, a situation very different from articulated skeletons in dedicated burials.

We have revised the description of the sampling material in Table S1.1 and Supplementary 1, indicating which tooth was sampled for each individual following international standardization (FDI).

“Testing for biological relatedness between the two individuals using the pairwise-mismatch rate (PMR) revealed unusually low values of 0.15246 (Supplementary 2, Table S1.5). After excluding four libraries from MLZ003 with signs of contamination, the PMR value was found to be even lower (0.1039)”

I would like to be much more confident that there is a sound statistical basis for combining the data from these two elements. Alternatively, is there potential to run the analyses with each separated to determine whether results are congruent, as one would expect from the same individual. Please also include more information on the two teeth.

We agree that estimating levels of kinship can be difficult when a study does not have a sufficiently large representation of unrelated individuals from which to estimate the extent of background relatedness in the population.

We have re-evaluated our findings: since all of the MLZ005 libraries were generated from extract(s) from the same skeletal element, and the same was done for the MLZ003 libraries, we are confident that all libraries came from the same individual. To compare the relatedness for the MLZ003 and MLZ005 libraries, we calculated PMR values, thinned for sites that are at least 500K sites apart for each pair, to avoid the effects of linkage disequilibrium.

We then compared the PMR values calculated on MLZ005/MLZ005 libraries (both libraries from the MLZ005 element), MLZ003/MLZ003 libraries (both libraries from the MLZ003 element) and MLZ003/MLZ005 libraries (one library from each element). Note that the ranked order of the PMR values is invariant to a normalizing constant, and so we can ignore the background relatedness, and use the raw PMR values.

We then performed pairwise-Wilcoxon rank sum tests and found no significant difference in the PMR values of the three library types (adjusted $p > 0.15$). However, to be more confident, we restricted PMR values to be based on at least 500 overlapping SNPs (removing the single MLZ003/MLZ003 case), performed an additional Wilcoxon rank sum test, and found no significant difference again ($p = 0.1974$).

Since we find no statistically significant difference between the distribution of the PMR values for combinations of libraries of MLZ003/MLZ003, MLZ003/MLZ005 and MLZ005/MLZ005, we have no evidence to suggest that MLZ003/MLZ005 libraries cannot be merged.

Finally, the fact that we obtained low levels of nuclear contamination after merging libraries from MLZ005 and MLZ003 also supports the assumption that both samples belong to the same individual/biological twins.

"The final coverage of MLZ003005 was 0.41X, corresponding to 226,914 autosomal SNPs in the 1240k panel"

I'd like to understand how this coverage was determined. Given the data is SNP capture, this 0.41x coverage isn't likely to be whole genome coverage. If so, how has this been calculated? Is this instead average coverage of all 1240k sites?

All coverage statistics are calculated "on-target" for the sites included in the 1240k SNP capture. Hence, we report the on-target coverage statistics for each of the libraries from MLZ003 and MLZ005 (reported in Table S1.2), but also report the overall on-target coverage of the sample of the merged libraries, denoted MLZ003005, as 0.41X. Note that the merged sample also covers 226,914 sites on the 1240k SNP panel.

Figure 4: "Results show that MLZ is the only Pleistocene HG that indicated a significant and equal attraction to both Natufian and Villabruna-like ancestries"

What about Paglicci133 individual in same figure? This would be easier to follow if Pleistocene HGs were defined in the text/figure.

We find that MLZ is one of the temporally oldest individuals that shows a positive attraction to both Natufians and Villabruna, which was observed by significantly positive f_4 -statistics (Z-score for Natufian test = 3.541, Z-score for Villabruna test = 2.972).

We emphasize the significance also in the caption, reading: "Results show that MLZ is the only Pleistocene HG that indicated a significant and equal attraction to both Natufian and Villabruna-like ancestries".

We have highlighted Pleistocene and Holocene individuals in Figure 4 now. Of note, we removed Paglicci133 from the analyses, as it was flagged as problematic (modern DNA contamination) in the latest data release version (<https://reich.hms.harvard.edu/allen-ancient->

dna-resource-aadr-downloadable-genotypes-present-day-and-ancient-dna-data).

Contamination from modern-day Europeans could indeed inflate the amount of shared drift with Natufian-like ancestries, as Near Eastern ancestry has increased with the arrival of the Neolithic and has persisted in Europe since then.

Great use of the PMD tools subsetted bam alongside the whole bam. However, could be clearer in the main text as to the extent of this subset (prop. of original/whole bam). See supplemental comments.

Thank you. We have added the following paragraph to make this point clearer (pg. 10, lines 258-263):

"Despite the absence of signs of contamination, all genomic analyses have been performed on the standard genotype data of MLZ003005 and the PMD-filtered data (MLZ003005.PMD). Additionally, in order to evaluate if DNA damage could have an effect on the results, we also genotyped MLZ003005 retaining only transversions (MLZ003005.TV) and replicated the analyses using this version."

Conclusions:

"Genome-wide data from the first Solutrean-associated individual MLZ in southern Iberia revealed a genetic link with an IUP population, which also connects IUP individuals Bacho Kiro, Tianyuan and Goyet Q116-1 from regions in today's Belgium, Bulgaria and China. The same ancestral population later split into two groups, which contributed to European and Asian populations, respectively"

Perhaps change "revealed a genetic link" to "is descended from". Clarify what is meant by "also connects".

Thanks. We have now changed that paragraph following the reviewer's suggestion to the following:

"Genome-wide data from the first Solutrean-associated individual MLZ in southern Iberia revealed traces of an IUP population that predates the genetic split between European and Asian populations. This genetic ancestry was also found in IUP individuals Bacho Kiro and Tianyuan as well as the Aurignacian individual Goyet Q116-1 from regions in today's Bulgaria, China and Belgium. Our results have shown that this IUP ancestry is still traceable to as late as 23,000 years BP in Southern Iberia."

Material and Methods:

Methods contemporary. As suggested, please add additional analysis to add certainty to sample combination.

Please see our comments about biological relatedness results provided above and the rationale for merging MLZ003 and MLZ005 libraries.

"Libraries with >0.1 % endogenous DNA, were subsequently enriched for ~1240k single nucleotide polymorphisms (SNPs)... "

In the results section the value was 0.08%

Corrected by "Libraries with ~0.1 % or higher endogenous DNA were subsequently...". Thanks for pointing this out.

Supplementals:

Capitalise "Francisco contreras" (?)

Done!

What type/form of tooth were sample ie (canines/molars etc)? Would also provide info to further support the combination of the two samples/

A precise description of the teeth that were sampled has now been added in the Supplementary information 1. We would like to add that this information was already available in Table S1.1 of the original submission.

2.2 PMD filtering section

Supplemental information incorrect: please check this.

Loss read % indicates the % reads retained, not removed. Therefore, ADS001 actually the best-preserved sample. Mix up in the % and sample name in the text here.

PMD score of below 3 doesn't directly correspond with contamination, but rather the degree of deamination present which is a predictor of contamination. Is contamination the correct word at all?

Thanks. "Loss reads %" was corrected to "Reads retained %". We have rephrased the paragraph describing the PMD filtering, highlighting that the reads which were not retained could have potentially come from a more modern contamination source.

Supplemental figure S1c

There is a slight error here ; PDM >>> PMD

Corrected, thank you!

Decision Letter, first revision:

7th December 2022

Dear Dr. Villalba-Mouco,

Thank you for submitting your revised manuscript "A 23,000-year-old southern-Iberian individual links human groups that lived in Western Europe before and after the Last Glacial Maximum" (NATECOLEVOL-220616880A). It has now been seen again by the original reviewers and their comments are below. The reviewers find that the paper has improved in revision, and therefore we'll be happy in principle to publish it in Nature Ecology & Evolution, pending minor revisions to satisfy the reviewers' final requests and to comply with our editorial and formatting guidelines.

[REDACTED]

Reviewer #1 (Remarks to the Author):

Thank you for sending me to review the manuscript again. I think most of the comments that I made in the first manuscript draft have been addressed. Indeed, the Introduction has been extended and a much clearer understanding of the state-of-the-art and previous publications is offered now. Moreover, all the archaeological remarks that I did in my first revision have been addressed.

I have a few new comments following the new sections/paragraphs to the paper that maybe the authors might want to take into consideration:

1/ "Some scholars have explained the 104 cultural discontinuity by migratory processes, with putative origins in North Africa on the basis of parallels with Aterian lithic assemblages (e.g.12,13 105". In this regard see also: Castaño, M.A., 2007. El Aterense del Norte de África y el Solutrense peninsular: ¿contactos transgibraltareños en el Pleistoceno Superior?. Munibe, 58, pp.101-126.

2/ "Cueva del Malalmuerzo is well known for its rock art 163 paintings that are stylistically attributed to the Solutrean. The latest archaeological 164 investigations of the cave uncovered several human remains in a small area, which 165 corresponded to an old archaeological profile from previous

25excavations. In parallel to 166 screening for ancient DNA, the remainder of two human teeth were radiocarbon dated, which 167 directly attributed the samples to the Solutrean period." The Solutrean is a technocomplex defined by its industry, not by dates. I have read the few publications on Cueva del Malalmuerzo. I think it would be appropriate to state clearly in the main manuscript that, even if there are Solutrean industries on the site (e.g. Garía Baba, C., Afonso Marrero, J.A., Martínez Fernandez, G., 1998. La modificación primaria en el proceso de la producción lítica. El caso de la producción laminar Solutrense de la Cueva de Malalmuerzo (Moclín, Granada). In: Sanchidrian Torti, J. L., Simon-Vallejo, M.D. (Eds.), Las Culturas del Pleistoceno Superior en Andalucía. Patronato de la Cueva de Nerja, Nerja, pp. 141–156.), so far there is no documentation of in situ stratigraphical layers associated with the Solutrean technocomplex. In fact, in Cabello, Lidia, et al. "New archaeological data on the upper Paleolithic site of cueva de Malalmuerzo (Moclín, Granada, Spain)." *Munibe Antropologia-Arkeologia* 71 (2020): 41-57, as the authors explain in the SI, the layers and the radiocarbon dating of the new excavations are attributed to the Magdalenian. In order to weigh up these human remains I think it is appropriate to at least mention that so far there are no in situ stratigraphical layers associated with this technocomplex. In the paper, big statements about the Solutrean are made for the whole of Western Europe. The reader needs to have this piece of information.

3/ "given that the Solutrean techno-complex is restricted to Southern 317 France and Iberia and that southwestern Europe was a geographical refugium for Upper 318 Paleolithic populations during the LGM, population continuity through time is a parsimonious 319 explanation". ...

"By contrast, Iberian hunter-gatherer groups carry 79 a genetic legacy that predates the LGM, which points to different dynamics in the proposed 80 southern refugia of Ice Age Europe and characterizes Iberia as the main refugium for Western 81 European pre-LGM ancestry"

I would be cautious stating (and taking for granted) that SW Europe/Iberia was a refugium. See for example this recent publication:

Canessa, T., 2021. Mobility and settlement strategies in southern Iberia during the Last Glacial Maximum: Evaluating the region's refugium status. *Journal of Archaeological Science: Reports*, 37, p.102966.

Related to this, in the conclusion is stated: "The genetic continuity suggests that the Iberian Peninsula was a southern 25 654 refugium that sustained a stable population before, during and after the LGM, with no evidence 655 for significant population turnover events but with an early and substantial contribution of 656 Villabruna-like HG ancestry".

Genetic continuity does not necessarily imply the existence of a refugium. There might be other complex scenarios/explanations that might have happened over such a long period of time.

4/ "as the 15 398 archeological record only provides evidence of IUP industries (e.g. Chatelperronian) in northern Iberia and broadly attributed to Late Neanderthals and not modern humans 58 399" . I would say Early Upper Paleolithic instead of IUP. Historiographically, at least in the Iberian Peninsula, that is the terminology used for Chatelperronian, Aurignacian, and Gravettian. IUP is terminology from the Middle East, and now Eastern Europe.

5/In the SI "The South of the Iberian Peninsula was considered to be a region where Late Neanderthals survived longest, coinciding with the Evolved Aurignacian^{54,55}. Finlayson et al. 56 56"

The first one to propose that late Neanderthals survived longer for Southern Iberia/Eastern Andalusia was L. Gerardo Vega Toscano. I think it would be fair that the authors reference his work for this particular statement.

...

I think all of these changes are quite easy to make as are nuances or small comments that can be easily added and I do not need to see the manuscript again. I trust the good criteria of the Editors and the authors.

Finally, my sincere apologies to the authors for the delay in this review. I have no excuses other than an overload of work.

Thank you to Nature Ecology and Evolution Journal and the authors for letting me do this review, I have enjoyed a lot the reading and the work, and last but not least I have learned a lot.

Paloma de la Peña

Reviewer #2 (Remarks to the Author):

The study by Villalba-Mouco et al., where genome-wide data from ancient humans on the Iberian Peninsula were compared to that of a wide variety of ancient Eurasians, reads very compellingly in this second iteration. With the re-writing and re-organization of key sections throughout the study, it is clear the impact of sequencing these ancient Iberian individuals, particularly the 23,000-year-old MLZ individual, filling in a much-needed gap in the ancestry in western Europe between the Aurignacian and Magdalenian periods. My comments below are very minor, detailed below:

I'm sure this will be resolved with any copy editing, but with the numerical references, there are odd sentence endings occasionally (e.g. L. 153, which ends with just 'after'). I'm assuming all of these cases are referring to a study, but worth checking in case there were missing pieces of a sentence. (also noticed at L. 201, 304).

On L. 166 and L. 172-177, citing Table S1.1 is useful.

In Figure 2a, it may be useful to also label El Miron, since you directly refer to this placement on the MDS figure in the text.

On L. 280, replace Magdalenian in the f4-statistic with Goyet Q2 cluster, to match the other f4-statistics in the text and in Figure 2b.

On L. 293-296, the language suggests MLZ is the ancestral lineage of the Goyet Q2 cluster. However, MLZ is an individual and may not be directly contributing to later individuals. Re-write to clarify this, perhaps by saying "Identifying MLZ as carrying ancestry associated with the lineage contributing to the Goyet Q2 cluster..."

On L. 348, add 'EXCESS genetic drift'

27On L. 352-355, the sentence is confusing. Perhaps reword as "Our results confirm that part of the IUP ancestry present in Bacho Kiro and Tianyuan also survived in the Southern Iberia MLZ until 23,000 BP, ~12,000 years later than the Aurignacian-associated Goyet Q116-1, the youngest previously known individual where this ancestry is traceable."

On L. 370, rewrite to 'than Bacho Kiro IUP shares with Tianyuan', to convey this was a comparison of two separate f4-statistics.

Figure 3B – Change 'test' to Old UP to match that in Fig. 3A

L. 534 – Give associated Z-score

L. 577 – I don't think listing all samples here is necessary, especially since you only use five individuals from this set for further genetic analyses.

Fig. S10 – add worst f-statistic to legend

Reviewer #3 (Remarks to the Author):

The authors have responded to our comments and made relevant changes to the manuscript. The justification of sample merging is robust and considered, and all relevant changes to the supplementary material have been made.

My only remaining question relates to the validity of the statement that "We have also observed consistently negative results using F4 statistics ... although these differences are not always significant."

The data only shows El Miron to share a significant excess affinity with the Villabruna Cluster, with all other samples having a Z score less than 3. A recent publication has shown that El Miron is admixed (<https://www.nature.com/articles/s41559-022-01883-z>) so an affinity to Villabruna is not surprising. No other GoyetQ2 associated individuals show such admixture, and from this study show no significant affinity to Villabruna. Could this be reconsidered in the manuscript?

Our ref: NATECOLEVOL-220616880A

12th December 2022

Dear Dr. Villalba-Mouco,

Thank you for your patience as we've prepared the guidelines for final submission of your Nature Ecology & Evolution manuscript, "A 23,000-year-old southern-Iberian individual links human groups that lived in Western Europe before and after the Last Glacial Maximum" (NATECOLEVOL-220616880A). Please carefully follow the step-by-step instructions provided in the attached file, and add a response in each row of the table to indicate the changes that you have made. Please also check and comment on any additional marked-up edits we have proposed within the text. Ensuring that each point is addressed will help to ensure that your revised manuscript can be swiftly handed over to our production team.

****We would like to start working on your revised paper, with all of the requested files and forms, as soon as possible (preferably within two weeks). Please get in contact with us immediately if you anticipate it taking more than two weeks to submit these revised files.****

In recognition of the time and expertise our reviewers provide to Nature Ecology & Evolution's editorial process, we would like to formally acknowledge their contribution to the external peer review of your manuscript entitled "A 23,000-year-old southern-Iberian individual links human groups that lived in Western Europe before and after the Last Glacial Maximum". For those reviewers who give their assent, we will be publishing their names alongside the published article.

Nature Ecology & Evolution offers a Transparent Peer Review option for new original research manuscripts submitted after December 1st, 2019. As part of this initiative, we encourage our authors to support increased transparency into the peer review process by agreeing to have the reviewer comments, author rebuttal letters, and editorial decision letters published as a Supplementary item. When you submit your final files please clearly state in your cover letter whether or not you would like to participate in this initiative. Please note that failure to state your preference will result in delays in accepting your manuscript for publication.

Cover suggestions

29As you prepare your final files we encourage you to consider whether you have any images or illustrations that may be appropriate for use on the cover of Nature Ecology & Evolution.

Nature Ecology & Evolution has now transitioned to a unified Rights Collection system which will allow our Author Services team to quickly and easily collect the rights and permissions required to publish your work. Approximately 10 days after your paper is formally accepted, you will receive an email in providing you with a link to complete the grant of rights. If your paper is eligible for Open Access, our Author Services team will also be in touch regarding any additional information that may be required to arrange payment for your article.

Please note that *Nature Ecology & Evolution* is a Transformative Journal (TJ). Authors may publish their research with us through the traditional subscription access route or make their paper immediately open access through payment of an article-processing charge (APC). Authors will not be required to make a final decision about access to their article until it has been accepted. [Find out more about Transformative Journals](https://www.springernature.com/gp/open-research/transformative-journals)

Authors may need to take specific actions to achieve [compliance](https://www.springernature.com/gp/open-research/funding/policy-compliance-faqs) with funder and institutional open access mandates. If your research is supported by a funder that requires immediate open access (e.g. according to [Plan S principles](https://www.springernature.com/gp/open-research/plan-s-compliance)) then you should select the gold OA route, and we will direct you to the compliant route where possible. For authors selecting the subscription publication route, the journal's standard licensing terms will need to be accepted, including [self-archiving-and-license-to-publish](https://www.nature.com/nature-portfolio/editorial-policies/self-archiving-and-license-to-publish). Those licensing terms will supersede any other terms that the author or any third party may assert apply to any version of the manuscript.

For information regarding our different publishing models please see our <https://www.springernature.com/gp/open-research/transformativ-journals> Transformativ Journals page. If you have any questions about costs, Open Access requirements, or our legal forms, please contact ASJournals@springernature.com.

Please use the following link for uploading these materials:
[REDACTED]

[REDACTED]

Reviewer #1:

Remarks to the Author:

Thank you for sending me to review the manuscript again. I think most of the comments that I made in the first manuscript draft have been addressed. Indeed, the Introduction has been extended and a much clearer understanding of the state-of-the-art and previous publications is offered now. Moreover, all the archaeological remarks that I did in my first revision have been addressed.

I have a few new comments following the new sections/paragraphs to the paper that maybe the authors might want to take into consideration:

1/ "Some scholars have explained the 104 cultural discontinuity by migratory processes, with putative origins in North Africa on the basis of parallels with Aterian lithic assemblages (e.g.12,13 105". In this regard see also: Castaño, M.A., 2007. El Aterense del Norte de África y el Solutrense peninsular:¿contactos transgibraltareños en el Pleistoceno Superior?. Munibe, 58, pp.101-126.

2/ "Cueva del Malalmuerzo is well known for its rock art 163 paintings that are stylistically attributed to the Solutrean. The latest archaeological 164 investigations of the cave uncovered several human remains in a small area, which 165 corresponded to an old archaeological profile from previous excavations. In parallel to 166 screening for ancient DNA, the remainder of two human teeth were radiocarbon dated, which 167 directly attributed the samples to the Solutrean period." The Solutrean is a technocomplex defined by its industry, not by dates. I have read the few publications on Cueva del Malalmuerzo. I think it would be appropriate to state clearly in the main manuscript that, even if there are Solutrean industries on the site (e.g. Garía Baba, C., Afonso Marrero, J.A., Martínez Fernandez, G., 1998. La modificación primaria en el proceso de la producción lítica. El caso de la producción laminar Solutrense de la Cueva de Malalmuerzo (Moclín, Granada). In: Sanchidrián Torti, J. L., Simon-Vallejo, M.D. (Eds.), Las Culturas del Pleistoceno Superior en Andalucía. Patronato de la Cueva de Nerja, Nerja, pp. 141–156.), so far there is no documentation of in situ stratigraphical layers associated with the Solutrean technocomplex. In fact, in Cabello, Lidia, et al. "New archaeological data on the upper Paleolithic site of cueva de Malalmuerzo (Moclín, Granada,

31Spain)." *Munibe Antropologia-Arkeologia* 71 (2020): 41-57, as the authors explain in the SI, the layers and the radiocarbon dating of the new excavations are attributed to the Magdalenian. In order to weigh up these human remains I think it is appropriate to at least mention that so far there are no in situ stratigraphical layers associated with this technocomplex. In the paper, big statements about the Solutrean are made for the whole of Western Europe. The reader needs to have this piece of information.

3/ "given that the Solutrean techno-complex is restricted to Southern 317 France and Iberia and that southwestern Europe was a geographical refugium for Upper 318 Paleolithic populations during the LGM, population continuity through time is a parsimonious 319 explanation". ...

"By contrast, Iberian hunter-gatherer groups carry 79 a genetic legacy that predates the LGM, which points to different dynamics in the proposed 80 southern refugia of Ice Age Europe and characterizes Iberia as the main refugium for Western 81 European pre-LGM ancestry"

I would be cautious stating (and taking for granted) that SW Europe/Iberia was a refugium. See for example this recent publication:

Canessa, T., 2021. Mobility and settlement strategies in southern Iberia during the Last Glacial Maximum: Evaluating the region's refugium status. *Journal of Archaeological Science: Reports*, 37, p.102966.

Related to this, in the conclusion is stated: "The genetic continuity suggests that the Iberian Peninsula was a southern 25 654 refugium that sustained a stable population before, during and after the LGM, with no evidence 655 for significant population turnover events but with an early and substantial contribution of 656 Villabruna-like HG ancestry".

Genetic continuity does not necessarily imply the existence of a refugium. There might be other complex scenarios/explanations that might have happened over such a long period of time.

4/ "as the 15 398 archeological record only provides evidence of IUP industries (e.g. Chatelperronian) in northern Iberia and broadly attributed to Late Neanderthals and not modern humans 58 399". I would say Early Upper Paleolithic instead of IUP. Historiographically, at least in the Iberian Peninsula, that is the terminology used for Chatelperronian, Aurignacian, and Gravettian. IUP is terminology from the Middle East, and now Eastern Europe.

5/In the SI "The South of the Iberian Peninsula was considered to be a region where Late Neanderthals survived longest, coinciding with the Evolved Aurignacian^{54,55}. Finlayson et al. 56 56"

The first one to propose that late Neanderthals survived longer for Southern Iberia/Eastern Andalusia was L. Gerardo Vega Toscano. I think it would be fair that the authors reference his work for this particular statement.

...

I think all of these changes are quite easy to make as are nuances or small comments that can be easily added and I do not need to see the manuscript again. I trust the good criteria of the Editors and the authors.

Finally, my sincere apologies to the authors for the delay in this review. I have no excuses other than an overload of work.

Thank you to Nature Ecology and Evolution Journal and the authors for letting me do this review, I have enjoyed a lot the reading and the work, and last but not least I have learned a lot.

Paloma de la Peña

Reviewer #2:

Remarks to the Author:

The study by Villalba-Mouco et al., where genome-wide data from ancient humans on the Iberian Peninsula were compared to that of a wide variety of ancient Eurasians, reads very compellingly in this second iteration. With the re-writing and re-organization of key sections throughout the study, it is clear the impact of sequencing these ancient Iberian individuals, particularly the 23,000-year-old MLZ individual, filling in a much-needed gap in the ancestry in western Europe between the Aurignacian and Magdalenian periods. My comments below are very minor, detailed below:

I'm sure this will be resolved with any copy editing, but with the numerical references, there are odd sentence endings occasionally (e.g. L. 153, which ends with just 'after'). I'm assuming all of these cases are referring to a study, but worth checking in case there were missing pieces of a sentence. (also noticed at L. 201, 304).

On L. 166 and L. 172-177, citing Table S1.1 is useful.

In Figure 2a, it may be useful to also label El Miron, since you directly refer to this placement on the MDS figure in the text.

On L. 280, replace Magadalenian in the f4-statistic with Goyet Q2 cluster, to match the other f4-statistics in the text and in Figure 2b.

On L. 293-296, the language suggests MLZ is the ancestral lineage of the Goyet Q2 cluster. However, MLZ is an individual and may not be directly contributing to later individuals. Re-write to clarify this, perhaps by saying "Identifying MLZ as carrying ancestry associated with the lineage contributing to the Goyet Q2 cluster..."

On L. 348, add 'EXCESS genetic drift'

On L. 352-355, the sentence is confusing. Perhaps reword as "Our results confirm that part of the IUP ancestry present in Bacho Kiro and Tianyuan also survived in the Southern Iberia MLZ until 23,000 BP, ~12,000 years later than the Aurignacian-associated Goyet Q116-1, the youngest previously known individual where this ancestry is traceable."

On L. 370, rewrite to 'than Bacho Kiro IUP shares with Tianyuan', to convey this was a comparison of two separate f4-statistics.

Figure 3B – Change 'test' to Old UP to match that in Fig. 3A

33L. 534 – Give associated Z-score

L. 577 – I don't think listing all samples here is necessary, especially since you only use five individuals from this set for further genetic analyses.

Fig. S10 – add worst f-statistic to legend

Reviewer #3:

Remarks to the Author:

The authors have responded to our comments and made relevant changes to the manuscript. The justification of sample merging is robust and considered, and all relevant changes to the supplementary material have been made.

My only remaining question relates to the validity of the statement that "We have also observed consistently negative results using F4 statistics ... although these differences are not always significant."

The data only shows El Miron to share a significant excess affinity with the Villabruna Cluster, with all other samples having a Z score less than 3. A recent publication has shown that El Miron is admixed (<https://www.nature.com/articles/s41559-022-01883-z>) so an affinity to Villabruna is not surprising. No other GoyetQ2 associated individuals show such admixture, and from this study show no significant affinity to Villabruna. Could this be reconsidered in the manuscript?

Author Rebuttal, first revision:Reviewer #1 (Remarks to the Author):

Thank you for sending me to review the manuscript again. I think most of the comments that I made in the first manuscript draft have been addressed. Indeed, the Introduction has been extended and a much clearer understanding of the state-of-the-art and previous publications is offered now. Moreover, all the archaeological remarks that I did in my first revision have been addressed.

I have a few new comments following the new sections/paragraphs to the paper that maybe the authors might want to take into consideration:

1/ "Some scholars have explained the 104 cultural discontinuity by migratory processes, with putative origins in North Africa on the basis of parallels with Aterian lithic assemblages (e.g.12,13 105".

In this regard see also: Castaño, M.A., 2007. El Aterense del Norte de África y el Solutrense peninsular: ¿contactos transgibraltareños en el Pleistoceno Superior?. Munibe, 58, pp.101-126.

Thanks for the pointer. The reference to Castaño 2007 has been added.

2/ "Cueva del Malalmuerzo is well known for its rock art 163 paintings that are stylistically attributed to the Solutrean. The latest archaeological 164 investigations of the cave uncovered several human remains in a small area, which 165 corresponded to an old archaeological profile from previous excavations. In parallel to 166 screening for ancient DNA, the remainder of two human teeth were radiocarbon dated, which 167 directly attributed the samples to the Solutrean period."

The Solutrean is a technocomplex defined by its industry, not by dates. I have read the few publications on Cueva del Malalmuerzo. I think it would be appropriate to state clearly in the main manuscript that, even if there are Solutrean industries on the site (e.g. Garía Baba, C., Afonso Marrero, J.A., Martínez Fernandez, G., 1998. La modificación primaria en el proceso de la producción lítica. El caso de la producción laminar Solutrense de la Cueva de Malalmuerzo (Moclín, Granada). In: Sanchidrian Torti, J. L., Simon-Vallejo, M.D. (Eds.), Las Culturas del Pleistoceno Superior en Andalucía. Patronato de la Cueva de Nerja, Nerja, pp. 141–156.), so far there is no documentation of *in situ* stratigraphical layers associated with the Solutrean technocomplex. In fact, in Cabello, Lidia, et al. "New archaeological data on the upper Paleolithic site of cueva de Malalmuerzo (Moclín, Granada, Spain)." Munibe Antropología-Arkeología 71 (2020): 41-57, as the authors explain in the SI, the layers and the radiocarbon dating of the new excavations are attributed to the Magdalenian.

In order to weigh up these human remains I think it is appropriate to at least mention that so far there are no *in situ* stratigraphical layers associated with this technocomplex. In the paper, big statements about the Solutrean are made for the whole of Western Europe. The reader needs to have this piece of information.

We have added the reference to Barba et al. 1998 and we clarify in the main text that so far there are not *in situ* stratigraphic layers associated with this technocomplex. Additionally, in order to make clear that the Solutrean is a technocomplex and not a time period, the last sentence of the paragraph now reads as follow:

"Samples MLZ003 and MLZ005 were found to be contemporaneous and radiocarbon dated to a period when the Solutrean techno-complex prevailed".

3/ "given that the Solutrean techno-complex is restricted to Southern 317 France and Iberia and that southwestern Europe was a geographical refugium for Upper 318 Paleolithic populations during the LGM, population continuity through time is a parsimonious 319 explanation". ...

"By contrast, Iberian hunter-gatherer groups carry 79 a genetic legacy that predates the LGM, which points to different dynamics in the proposed 80 southern refugia of Ice Age Europe and characterizes Iberia as the main refugium for Western 81 European pre-LGM ancestry"

I would be cautious stating (and taking for granted) that SW Europe/Iberia was a refugium. See for example this recent publication:

Canessa, T., 2021. Mobility and settlement strategies in southern Iberia during the Last Glacial Maximum: Evaluating the region's refugium status. *Journal of Archaeological Science: Reports*, 37, p.102966.

Related to this, in the conclusion is stated: "The genetic continuity suggests that the Iberian Peninsula was a southern 25 654 refugium that sustained a stable population before, during and after the LGM, with no evidence 655 for significant population turnover events but with an early and substantial contribution of 656 Villabruna-like HG ancestry".

Genetic continuity does not necessarily imply the existence of a refugium. There might be other complex scenarios/explanations that might have happened over such a long period of time.

Thanks. We have read the publication by Canessa 2021 carefully. This article focuses on the role of Southern Iberia as an ecological refugium based on the mobility of human groups, differentiated according to retouched tool assemblages within a specific area of Iberia.

Canessa 2021 write in the discussion:

"A significant consequence of this perspective is that the classic and widely accepted model which links regional settlement to discrete ecological refugia appears as an inadequate simplification (Straus et al., 2000b). Crucially, this reinforces the complex scenario of settlement and human-environment interactions emphasised by the growing discovery of sites from interior and so-called inhospitable regions of Iberia (e.g. Fernández Gómez & Velasco Ortiz, 2013; Alcaraz-Castaño, 2015; Yravedra et al., 2016; Alcaraz-Castaño et al., 2017)."

Our results do not contradict the interaction of groups within and between regions of Iberia, as Canessa 2021 concludes. However, in our paper we refer to Iberia as a genetic refugium, and this is not limited to Southern Iberia, but in fact concerns the observation of a long-term genetic stability in Western Europe. This situation is different from the genetic transformations in Central Europe or the Italian Peninsula and led us to conclude that the Iberian Peninsula must have provided a range of suitable habitats that could sustain stable population(s) in and through which pre-LGM ancestry survived.

We agree with the reviewer that genetic continuity might not necessarily imply the existence of an **ecological** refugium (the term to which Canessa refers). Unfortunately, the scarcity of genomic data from this time period does permit exploration of more complex scenarios (yet). We therefore rephrased the respective sentences in the abstract and the conclusion, adding more caution to the interpretation.

4/ "as the 15 398 archeological record only provides evidence of IUP industries (e.g. Chatelperronian) in northern Iberia and broadly attributed to Late Neanderthals and not modern humans 58 399" . I would say Early Upper Paleolithic instead of IUP. Historiographically, at least in the Iberian Peninsula, that is the terminology used for Chatelperronian, Aurignacian, and Gravettian. IUP is terminology from the Middle East, and now Eastern Europe.

Thanks for pointing this out. We have changed IUP to Early UP now.

5/In the SI "The South of the Iberian Peninsula was considered to be a region where Late Neanderthals survived longest, coinciding with the Evolved Aurignacian^{54,55}. Finlayson et al. 56 56"

The first one to propose that late Neanderthals survived longer for Southern Iberia/Eastern Andalusia was L. Gerardo Vega Toscano. I think it would be fair that the authors reference his work for this particular statement.

The reference is added now, thanks!

I think all of these changes are quite easy to make as are nuances or small comments that can be easily added and I do not need to see the manuscript again. I trust the good criteria of the Editors and the authors.

Finally, my sincere apologies to the authors for the delay in this review. I have no excuses other than an overload of work.

Thank you to Nature Ecology and Evolution Journal and the authors for letting me do this review, I have enjoyed a lot the reading and the work, and last but not least I have learned a lot.

Paloma de la Peña

Thank you so much for the careful revision and the positive feedback.

Reviewer #2 (Remarks to the Author):

The study by Villalba-Mouco et al., where genome-wide data from ancient humans on the Iberian Peninsula were compared to that of a wide variety of ancient Eurasians, reads very compellingly in this second iteration. With the re-writing and re-organization of key sections throughout the study, it is clear the impact of sequencing these ancient Iberian individuals, particularly the 23,000-year-old MLZ individual, filling in a much-needed gap in the ancestry in western Europe between the Aurignacian and Magdalenian periods. My comments below are very minor, detailed below:

Thank you very much.

I'm sure this will be resolved with any copy editing, but with the numerical references, there are odd sentence endings occasionally (e.g. L. 153, which ends with just 'after'). I'm assuming all of these cases are referring to a study, but worth checking in case there were missing pieces of a sentence. (also noticed at L. 201, 304).

Yes, these were all referring to a study, which indeed looks odd when citations are turned into numbers. We thus removed the linking words.

On L. 166 and L. 172-177, citing Table S1.1 is useful.

Added

In Figure 2a, it may be useful to also label El Miron, since you directly refer to this placement on the MDS figure in the text.

Added.

On L. 280, replace Magadalenian in the f4-statistic with Goyet Q2 cluster, to match the other f4-statistics in the text and in Figure 2b.

Corrected.

On L. 293-296, the language suggests MLZ is the ancestral lineage of the Goyet Q2 cluster. However, MLZ is an individual and may not be directly contributing to later individuals. Rewrite to clarify this, perhaps by saying "Identifying MLZ as carrying ancestry associated with the lineage contributing to the Goyet Q2 cluster..."

Corrected.

On L. 348, add 'EXCESS genetic drift'

Corrected.

On L. 352-355, the sentence is confusing. Perhaps reword as "Our results confirm that part of the IUP ancestry present in Bacho Kiro and Tianyuan also survived in the Southern Iberia MLZ until 23,000 BP, ~12,000 years later than the Aurignacian-associated Goyet Q116-1, the youngest previously known individual where this ancestry is traceable."

Corrected.

On L. 370, rewrite to 'than Bacho Kiro IUP shares with Tianyuan', to convey this was a comparison of two separate f4-statistics.

Corrected.

Figure 3B – Change 'test' to Old UP to match that in Fig. 3A

We have changed Old UP to 'test' as Old UP does not correspond to any specific archaeological category. Figure 3A and Figure 3B are now consistent.

L. 534 – Give associated Z-score

Z-score added.

L. 577 – I don't think listing all samples here is necessary, especially since you only use five individuals from this set for further genetic analyses.

Numbers are removed now.

Fig. S10 – add worst f-statistic to legend

Worst Z-score added

Reviewer #3 (Remarks to the Author):

The authors have responded to our comments and made relevant changes to the manuscript. The justification of sample merging is robust and considered, and all relevant changes to the supplementary material have been made.

My only remaining question relates to the validity of the statement that "We have also observed consistently negative results using F4 statistics ... although these differences are not always significant."

This sentence refers to the results obtained by f_4 -statistics of the form $f_4(\text{MLZ, Goyet Q2 cluster; Villabruna, Mbuti})$. Here, depending on the proportion of Villabruna-like ancestry present in the tested individuals of the Goyet Q2 cluster (and/or the coverage of the individual tested), the results are significant or not, however, all of them returned negative f_4 -statistics. We have rephrased this paragraph to make it clearer, now reading:

"We also observed consistently negative results using f_4 -statistics of the form $f_4(\text{MLZ, Goyet Q2 cluster; Villabruna, Mbuti})$ with Z-scores ranging from -9.568 (El Mirón) to -0.091 (Hohle Fels), which suggests excess affinity to Villabruna-like ancestry in individuals from the Goyet Q2 cluster when compared to MLZ, even though the f_4 -statistics are not always significant ($|Z| > 3$) (Table S2.9)."

The data only shows El Miron to share a significant excess affinity with the Villabruna Cluster, with all other samples having a Z score less than 3. A recent publication has shown that El Miron is admixed (<https://www.nature.com/articles/s41559-022-01883-z>) so an affinity to Villabruna is not surprising. No other GoyetQ2 associated individuals show such admixture, and from this study show no significant affinity to Villabruna. Could this be reconsidered in the manuscript?

Of note we had already described Villabruna-like admixture in El Mirón in Villalba-Mouco et al. 2019, an observation which formed the rationale for additional tests in this study. Charlton et al. 2022 recently confirmed this, too, when discussing Goyet Q2 cluster.

In Villalba-Mouco et al. 2019, we decided to rename the El Miron cluster to Goyet Q2 cluster, because El Miron did not represent well genetically the rest of the Magdalenian-associated individuals well due to this high admixture levels with an ancestry related to Villabruna. At the time, Goyet Q2 was the second Magdalenian-associated individual with higher coverage, and following the naming criteria suggested by Fu et al., 2016 criteria, the cluster was renamed to Goyet Q2.

Please, also note the complementary tests that allowed us to detect Villabruna-like ancestry in all individuals belonging to the Goyet Q2 cluster, albeit at a lower proportion than in the El Mirón individual:

Using $f_4(\text{Goyet Q2 cluster}, \text{Goyet Q116-1}; \text{Villabruna}, \text{Mbuti})$ results in a Z-scores >3 for all individuals belonging to the Goyet Q2 cluster, except for Burkhardtshohle, which is the individual with the lowest coverage and thus underpowered (Table S2.9).

Using $f_4(\text{Goyet Q2 cluster}, \text{Kostenki 14}; \text{Villabruna}, \text{Mbuti})$, also results in Z-scores >3 for all individuals belonging to the Goyet Q2 (Table S.10 and Figure 4a). A Z score = 2.972 for MLZ in the same test close to the (arbitrary) threshold of significance is strongly suggestive of the presence of Villabruna-like ancestry in MLZ, but to a lower extent than in E Miron.

With regards to the potentially contradicting results reported in Charlton et al. 2022: If Goyet Q2 is used as a source of admixture in a qpAdm model it seems that it is possible to model the remainder of the non-Iberian, Magdalenian-associated individuals as 100% Goyet Q2 ancestry. However this is not direct evidence for the absence of Villabruna-like ancestry in individuals of the Goyet Q2-cluster, unless the tested individuals lacked an excess of Villabruna-like ancestry not present in the Goyet Q2 individual.

Final Decision Letter:

3rd January 2023

Dear Vanessa,

We are pleased to inform you that your Article entitled "A 23,000-year-old southern-Iberian individual links human groups that lived in Western Europe before and after the Last Glacial Maximum", has now been accepted for publication in Nature Ecology & Evolution.

Over the next few weeks, your paper will be copyedited to ensure that it conforms to Nature Ecology and Evolution style. Once your paper is typeset, you will receive an email with a link to choose the appropriate publishing options for your paper and our Author Services team will be in touch regarding any additional information that may be required

You will not receive your proofs until the publishing agreement has been received through our system

Due to the importance of these deadlines, we ask you please us know now whether you will be difficult to contact over the next month. If this is the case, we ask you provide us with the contact information (email, phone and fax) of someone who will be able to check the proofs on your behalf, and who will be available to address any last-minute problems . Once your paper has been scheduled for online publication, the Nature press office will be in touch to confirm the details.

Acceptance of your manuscript is conditional on all authors' agreement with our publication policies (see www.nature.com/authors/policies/index.html). In particular your manuscript must not be published elsewhere and there must be no announcement of the work to any media outlet until the publication date (the day on which it is uploaded onto our web site).

Please note that *Nature Ecology & Evolution* is a Transformative Journal (TJ). Authors may publish their research with us through the traditional subscription access route or make their paper immediately open access through payment of an article-processing charge (APC). Authors will not be required to make a final decision about access to their article until it has been accepted. [Find out more about Transformative Journals](https://www.springernature.com/gp/open-research/transformative-journals)

Authors may need to take specific actions to achieve [compliance with funder and institutional open access mandates](https://www.springernature.com/gp/open-research/funding/policy-compliance-faqs). If your research is supported by a funder that requires immediate open access (e.g. according to FAIR principles)

42[Plan S principles](https://www.springernature.com/gp/open-research/plan-s-compliance)) then you should select the gold OA route, and we will direct you to the compliant route where possible. For authors selecting the subscription publication route, the journal's standard licensing terms will need to be accepted, including <https://www.nature.com/nature-portfolio/editorial-policies/self-archiving-and-license-to-publish>. Those licensing terms will supersede any other terms that the author or any third party may assert apply to any version of the manuscript.

We welcome the submission of potential cover material (including a short caption of around 40 words) related to your manuscript; suggestions should be sent to Nature Ecology & Evolution as electronic files (the image should be 300 dpi at 210 x 297 mm in either TIFF or JPEG format). Please note that such pictures should be selected more for their aesthetic appeal than for their scientific content, and that colour images work better than black and white or grayscale images. Please do not try to design a cover with the Nature Ecology & Evolution logo etc., and please do not submit composites of images related to your work. I am sure you will understand that we cannot make any promise as to whether any of your suggestions might be selected for the cover of the journal.

You can generate the link yourself when you receive your article DOI by entering it here: <http://authors.springernature.com/share>.

[REDACTED]

P.S. Click on the following link if you would like to recommend Nature Ecology & Evolution to your librarian <http://www.nature.com/subscriptions/recommend.html#forms>

** Visit the Springer Nature Editorial and Publishing website at http://editorial-jobs.springernature.com?utm_source=ejp_NEcoE_email&utm_medium=ejp_NEcoE_email&utm_campaign=ejp_NEcoE for more information about our career opportunities. If you have any questions please click [here](mailto:editorial.publishing.jobs@springernature.com).**